
# How can observational data be used to improve the modeling of human-managed reservoirs in large-scale hydrological models?

Seyed-Mohammad Hosseini-Moghari[1], Petra Döll[1,2]

[1]Institute of Physical Geography, Goethe University Frankfurt, Frankfurt am Main, Germany

[2]Senckenberg Leibniz Biodiversity and Climate Research Centre Frankfurt (SBiK-F), Frankfurt am Main, Germany

*Correspondence to:* Seyed-Mohammad Hosseini-Moghari (hosseinimoghari@em.uni-frankfurt.de)

**Abstract.** Human-managed reservoirs alter water flows and storage, impacting the hydrological cycle. However, modeling reservoir outflow and storage is challenging because it depends on human decisions, and there is often limited access to data on inflows, outflows, storage, or operational rules. Consequently, large-scale hydrological models either exclude reservoir

operations or use calibration-free algorithms for modeling reservoir dynamics. Nowadays, remotely-sensed information on reservoir storage anomalies is a potential source for calibrating reservoir operation algorithms. However, it is not yet clear what impact calibration against storage anomalies has on simulated reservoir outflow and absolute storage. In this study, we introduce two reservoir operation algorithms that require calibration: the Scaling Algorithm (SA) and the Weighting Algorithm (WA). These algorithms were implemented in the global hydrological model WaterGAP and compared with the widely used Hanasaki

algorithm with both default (DH) and calibrated (CH) parameter values. We calibrated all three algorithms against outflow, storage, storage anomalies, and estimated storage (based on storage changes and reservoir capacity) observed for 100 reservoirs in the USA to understand the information content of the observation variables. As expected, calibration against all three types of storage-related variables improved the storage simulation. Storage simulation using DH resulted in only 16 (15) skillful simulations (where the Kling–Gupta Efficiency with a trend component > -0.73) out of 100 reservoirs. In contrast, calibration

against storage anomalies resulted in 64 (39), 68 (45), and 66 (45) skillful storage simulations for CH, SA, and WA, respectively, during the calibration (validation) period. However, calibration against storage-related variables barely improved the performance of the outflow simulation, which strongly depends on the accuracy of the simulated inflow. In fact, using observed inflow instead of simulated inflow has a more significant effect on improving outflow simulation than calibration, whereas the opposite is true for storage simulation. We found that the default parameters of the Hanasaki algorithm rarely matched the

calibrated parameters, highlighting the benefit of calibration. Moreover, taking into downstream water demand in the reservoir operation algorithm does not necessarily improve modeling performance due to high uncertainty in demand estimation. Overall, the SA algorithm outperforms the other algorithms. Therefore, to improve the modeling of reservoir storage and outflow, we recommend calibrating the SA reservoir operation algorithm against remote sensing-based storage anomalies and improving reservoir inflow simulation.

## 1   Introduction

Globally, more than 58,000 large dams (at least 15 meters in height), capable of impounding 8300 km³, have been constructed to meet various human needs such as irrigation, flood control, hydropower generation, domestic water supply, and recreation (Chao et al., 2008; Perera et al., 2021). These dams annually store about one-sixth of the streamflow in reservoirs (Hanasaki et al., 2006), significantly altering the global freshwater system by increasing evaporation and modifying downstream streamflow

(Best, 2019; Tian et al., 2022). About 60% of the seasonal variability in Earth's surface water storage is attributed to human-





managed reservoirs, i.e. artificial reservoirs and regulated lakes, as the water level of reservoirs varies on average four times as much as that of natural lakes (Cooley et al., 2021). Therefore, to accurately depict the hydrologic cycle, the inclusion of human-managed reservoirs in hydrological models is crucial. This inclusion is supposed to enhance model performance, particularly regarding evapotranspiration and streamflow. At present, six out of the 16 global hydrological models contributing to ISIMIP2

(The Inter-Sectoral Impact Model Intercomparison Project, www.isimip.org) simulate the dynamics of human-managed reservoirs (Telteu et al., 2021).

Whereas the outflow from a natural lake strongly depends on the water level of the lake and thus the water storage in the lake, humans control the outflow from a reservoir. Even though human decisions on the release of water from reservoirs do, to some degree, depend on reservoir water storage, they are also influenced by many other factors, such as downstream water

demand, the demand for hydropower production, the need to protect downstream regions from flooding, ecosystem requirements, and legal constraints (Jager and Smith, 2008; Dong et al., 2023). Most reservoirs serve multiple purposes, making their simulation even more complex. However, since reservoir operation rules and observations of reservoir inflow, outflow and storage dynamics are rarely publicly accessible, large-scale hydrological models need to resort to calibration-free reservoir operation algorithms that only require information about the reservoir's storage capacity and surface water area. They are calibration-free algorithms

in the sense that they do not require the calibration of reservoir-specific algorithm parameters based on observations of model output variables. These calibration-free algorithms can only very roughly simulate the decisions of reservoir operators and cannot account for the unique operation patterns of each reservoir (Masaki et al., 2018; Turner et al., 2021; Steyaert and Condon 2024).

All global hydrological models currently use calibration-free reservoir operation algorithms, which differ regarding their formulation and complexity (Telteu et al., 2021). Examples for calibration-free reservoir operation algorithms proposed for large-

scale hydrological modeling are described in Dong et al. (2022), Zajac et al. (2017), Haddeland et al. (2006), and Hanasaki et al. (2006) (herein referred to as H06). Dong et al. (2022) and Zajac et al. (2017) employed different operation rules for four distinct levels of reservoir storage in their algorithms, whereas Haddeland et al. (2006) developed a prospective optimization algorithm based on the reservoir purpose. The H06 method is currently implemented in the global hydrological model H08 (Hanasaki et al., 2008) and, in a slightly modified form, in the global hydrological model WaterGAP, and also serves as the

foundation for the Dam-Reservoir Operation model (DROP; Sadki et al., 2023). While studies (e.g., Döll et al., 2009; Vanderkelen et al., 2022) clearly demonstrate that implementing the H06 algorithm leads to improved streamflow simulations compared to not considering the reservoir as a surface water body at all, there is no consensus (please refer to Döll et al., 2009; Vanderkelen et al., 2022; Gutenson et al., 2020) on whether the H06 algorithm outperforms the natural lake outflow parameterization of Döll et al. (2003) (herein referred to as D03), which assumes artificial reservoirs behave similarly to natural

lakes. It should be noted that simulating reservoir outflow and storage dynamics depends not only on the reservoir operation algorithm but also on the quality of the simulated inflow, making it difficult to assess the adequacy of the algorithm without inflow observations (Vanderkelen et al., 2020).

Several studies have endeavored to fine-tune calibration-free algorithms by adjusting a single parameter for each reservoir, but the results have been unpromising. For example, Gutenson et al. (2020) found that adjusting only one parameter of H06 for

60 non-irrigation reservoirs across the US did not lead to better simulations compared to a calibrated D03. Shin et al. (2019) reported that a new algorithm based on H06, where one parameter was calibrated for 27 reservoirs, could not accurately capture the seasonality in reservoir storage and outflow. Consequently, some studies have devised calibration-required algorithms with multiple parameters for each reservoir. Turner et al. (2021) introduced the Inferred Storage Targets and Release Functions (ISTARF) approach, a reservoir operating policy with 19 parameters. This approach was applied to 1,930 reservoirs across the





US and demonstrated robust improvements in both outflow and storage compared to the H06 model. Although the ISTARF
approach is relatively parsimonious in terms of the number of parameters compared to other established calibration-required
algorithms — such as those proposed by Yassin et al. (2019) and Turner et al. (2020), which feature 72 (six parameters for each
month) and 208 parameters per reservoir (four parameters for each week), respectively — the integration of these approaches
into large-scale models incurs substantial computational expenses. More importantly, this approach requires time series data of

observed inflow, outflow, and reservoir storage, which can be difficult to obtain outside the US, rendering it infeasible for global-
scale modeling. The same limitation applies to some machine learning approaches for simulating reservoir dynamics, such as the
artificial neural network approach proposed by Ehsani et al. (2016) and the tree-based reservoir model of Chen et al. (2022).

Remotely sensed data on water levels and surface water area of reservoirs are increasingly available and are being used to
derive time series of water storage anomalies or even absolute storage. With recent advancements in spaceborne data, such as

the Surface Water and Ocean Topography (SWOT) mission, storage anomalies data can now be gathered even for small
reservoirs, providing a valuable source for enhancing resource modeling within large-scale hydrological models (Biancamaria et
al., 2016). Examples include HydroSat (Tourian et al., 2022), the Global Reservoir Storage (GRS) dataset (Li et al., 2023), and
GloLakes (Hou et al., 2024). This newly available information could be used to calibrate reservoir operation algorithms
individually for each reservoir, which is expected to lead to an improved simulation of reservoir dynamics. Remote sensing-

derived reservoir storage anomalies were shown to fit reasonably well to in-situ observations, depending on the reservoir and
satellite data product; storage anomalies rather than absolute water storage values should be considered for both the simulated
and remote sensing data (Otta et al., 2023). Dong et al. (2023) demonstrated that simultaneous calibrations against reconstructed
release and reservoir storage data (using remotely sensed data, model simulations, and in-situ data) considerably improved the
performance of reservoir operation algorithms for the Ertan and Jinping I reservoirs in China. However, for global-scale studies,

release information is unavailable for most reservoirs. In such cases, calibrating against storage anomalies alone for parameter
estimation may degrade outflow simulations due to potential trade-offs between calibrating against different variables (Döll et
al., 2024). The recently published dataset of observed dynamics of US reservoirs, 'ResOpsUS' (Steyaert et al., 2022), which
provides time series of daily observed storage, elevation, inflows, and outflows for up to 679 reservoirs across the contiguous
US, offers an opportunity to explore this trade-off.

The main objective of our study is to determine how time series of observed reservoir water storage anomalies are best used
to improve the simulation of both reservoir outflow and storage in continental or global hydrological models. Additionally, we
aim to quantify the extent to which such calibration can enhance model performance. To better understand the value of the more
widely available remote-sensing-based observations of water storage anomalies, we compared their value to the value of more
scarce outflow and water storage observations for model calibration. To achieve this, we calibrated three different reservoir

operation algorithms — H06, and two newly developed storage-based algorithms, the Scaling Algorithm (SA) and the Weighting
Algorithm (WA) — which were implemented in the global hydrological model WaterGAP 2.2e (Müller Schmied et al., 2023a),
using monthly time series of observed outflow and storage data for 100 reservoirs in the US. The in-situ data, sourced from the
ResOpsUS dataset, were used as the basis for assessing the performance of all four algorithms (SA, WA, calibrated H06 CH,
and uncalibrated H06 DH) regarding outflow, storage anomaly, storage, and estimated storage, where calibration was performed

using alternatively one of the four variables for calibration of the algorithms. In addition to evaluating the algorithms with
simulated inflow data from WaterGAP, we also conducted assessments based on observed inflow data for the 35 (out of the 100)
reservoirs with available inflow measurements. This was done to examine the potential sensitivity of the algorithms to the quality





of inflow data, as reported by Vanderkelen et al. (2022). Finally, we assessed the impact of considering versus not considering downstream demand for irrigation and supply reservoirs (21 reservoirs) on the performance of the reservoir operation algorithms.

## 2   Methods and Data

### 2.1 The global hydrological model WaterGAP

WaterGAP simulates the dynamics of water flows and storages on the continents as impacted by human water use and human-managed reservoirs (Müller Schmied et al., 2021). It computes sectoral water abstractions as well as net abstractions (abstraction minus return flows) from surface water bodies (reservoirs, lakes, and rivers) and from groundwater. The model has a spatial resolution of 0.5°×0.5° and a daily temporal resolution. However, the model output analysis is normally done at the monthly scale. The current version, 2.2e, has been calibrated in a basin-specific manner against the mean annual streamflow at 1,509 gauging stations worldwide (Müller Schmied et al., 2023a). Taking into account the commissioning years, WaterGAP simulates the dynamics of reservoirs with a storage capacity of at least 0.5 km$^3$, referred to as 'global' reservoirs, using a slightly adapted version of the H06 algorithm (Döll et al., 2009). Smaller reservoirs (termed "local" reservoirs) are treated as natural lakes (Müller Schmied et al., 2021). A total of 1,255 global reservoirs, with a combined maximum capacity of 5,672 km$^3$, are integrated into WaterGAP 2.2e, sourced from the GRanD (Lehner et al., 2011) and GeoDAR (Wang et al., 2022) datasets; in addition, 88 regulated lakes are treated like global reservoirs (Müller Schmied et al., 2023a). The water balance for a reservoir in WaterGAP is calculated as (Müller Schmied et al., 2021):

$$\frac{dS}{dt} = I + A \cdot \left( P - E_{pot} \right) - GWR - NAs - O \tag{1}$$

where $S$ (m$^3$) represents reservoir storage, $I$ (m$^3$/d) denotes inflow into the reservoir from upstream, $A$ (m$^2$) is the reservoir area, $P$ (m/d) indicates precipitation, $E_{pot}$ (m/d) stands for potential evaporation, $GWR$ (m$^3$/d) denotes groundwater recharge (only in arid/semiarid regions), $NAs$ (m$^3$/d) represents potential net abstraction from the reservoir, and $O$ (m$^3$/d) is the reservoir outflow including release and spill. The surface area $A$ is computed daily as a fraction of the maximum area that depends on the current reservoir storage and its storage capacity. $A$ is reduced by 15 % when $S$ reaches 50% of the reservoir's capacity, and by 75% when $S$ drops to 10% of the capacity (Müller Schmied et al., 2021). Abstraction from a reservoir is permitted only until the water storage level drops to 10% of its total capacity. The implementation of reservoir operation algorithms in WaterGAP is described below. For detailed information on WaterGAP, please refer to Müller Schmied et al. (2021, 2023a).

### 2.2 Reservoir operation algorithms

#### 2.2.1 Hanasaki algorithm as implemented in WaterGAP2.2e

The calibration-free H06 method, in its original formulation, estimates monthly reservoir outflow distinguishing irrigation and non-irrigation reservoirs. For non-irrigation reservoirs, this outflow is determined by factors such as the storage at the beginning of the operational year (determined by analyzing the seasonal flow dynamics), the mean annual inflow into the reservoir, and the reservoir storage capacity. The long-term target for reservoir releases is the mean annual inflow. If reservoir storage at the beginning of an operational year is above normal, releases are increased throughout the year, and if it is below normal, releases are decreased. Therefore, the total release in an operational year depends on the storage level at the start of that year. In the case of irrigation reservoirs, the demand also influences the outflow (Hanasaki et al., 2006). The H06 algorithm was implemented in WaterGAP on a daily time scale, and the mean annual inflow was adjusted by adding the difference between precipitation and





evaporation over the reservoir. This modification aimed to provide a more accurate representation of the reservoir's water balance (Döll et al., 2009).

The first step in the H06 algorithm involves determining the release coefficient for the operational year 'y' ($k_y$) using the following equation:

$$k_y = \frac{S_{ini}}{a_1 \cdot C} \tag{2}$$

where $S_{ini}$ (km³) represents the reservoir storage at the start of the operational year; $C$ (km³) denotes the water storage capacity of the reservoir; and $a_1$ is a parameter of the H06 method, recommended to be set to 0.85 in its standard form. In the second step, the provisional release is determined. For non-irrigation reservoirs, the provisional release is calculated as follows:

$$R'_d = \overline{I'} \tag{3}$$

in which $R'_d$ (m³/s) is the provisional release for the day 'd' and $\overline{I'}$ (m³/s) is the mean annual inflow into the reservoir plus the difference between precipitation and evaporation over the reservoir (for this study, the period 1980-2009). For irrigation reservoirs, the provisional release is computed as follows:

$$R'_d = \begin{cases} a_2 \cdot \overline{I'} \cdot \left[1 + \dfrac{k_{alc} \cdot NAs_d}{\overline{NAs}}\right] & if \ \overline{NAs} \geq a_2 \cdot \overline{I'} \\ \overline{I'} + k_{alc} \cdot NAs_d - \overline{NAs} & otherwise \end{cases} \tag{4}$$

in which $NAs_d$ (m³/s) represents the potential net abstraction from surface water bodies for downstream cells of the reservoir for day 'd'; $\overline{NAs}$ (m³/s) denotes the mean total annual potential net abstraction for downstream cells of the irrigation reservoir; $k_{alc}$ is an allocation coefficient that distributes the abstraction to the upstream reservoirs based on the proportion of $\overline{I'}$ into each reservoir (it equals one if there is only one irrigation reservoir upstream of the demand cells); and $a_2$ is a parameter specifically for irrigation reservoirs that acts as a partitioner, leading to the use of different equations for reservoirs with a high demand-to-inflow ratio compared to those with a low demand-to-inflow ratio. With a default value of 0.5, this parameter sets the minimum provisional release at 50% of the mean annual inflow during non-crop months. During crop months, the fluctuations in provisional release for reservoirs with a high demand-to-inflow ratio ($\overline{NAs}$ exceeding 50% of mean annual inflow, first equation) correspond to fluctuations in daily net abstraction relative to $\overline{NAs}$. In contrast, reservoirs with a low demand-to-inflow ratio (second equation) align their provisional releases with the daily net abstraction (Hanasaki et al., 2006). The downstream potential net abstraction associated with each reservoir is calculated based on surface water demand for a maximum of five grid cells downstream in the absence of other reservoirs. Otherwise, it extends to the next reservoir. The potential net abstraction information is obtained from the WaterGAP dataset.

With the provisional release determined, the daily release is calculated using the following equation:

$$R_d = \begin{cases} k_y \cdot R'_d & if \ c \geq a_3 \\ \left(\dfrac{c}{a_3}\right)^2 \cdot k_y \cdot R'_d + \left\{1 - \left(\dfrac{c}{a_3}\right)^2\right\} \cdot I_d & otherwise \end{cases} \tag{5}$$

where $c$ represents the ratio of $C$ (km³) to $\overline{I'}$ (km³/yr); $I_d$ (m³/s) is the daily inflow into the reservoir for the day 'd'; $R_d$ (m³/s) is the daily release from the reservoir; and $a_3$ is a third parameter in the H06 approach, with default value of 0.5. This parameter is also a partitioner that results in the application of different equations for reservoirs with high capacity-to-inflow ratios ($c \geq a_3$) compared to those with low capacity-to-inflow ratios. This implies that for reservoirs with high capacity-to-inflow ratios (first equation), release is independent of daily inflow, while for reservoirs with low capacity-to-inflow ratios (second equation), daily





inflow influences the release (Hanasaki et al., 2006). In this study, H06 with default values for $a_1$, $a_2$, and $a_3$ is referred to as the DH algorithm, while H06 with calibrated parameters is referred to as the CH algorithm.

### 2.2.2 New algorithms

In this study, we introduce and compare two reservoir operation algorithms that 1) require the reservoir-specific calibration of

their parameters and 2) different from H06, compute daily release as a function of daily reservoir water storage. Both models include three parameters that are related to different levels of storage: above 70% of the reservoir capacity (level 1), between 40% and 70% of the reservoir capacity (level 2), and below 40% of the reservoir capacity (level 3). This classification is based on the observation that the operation rule curve of reservoirs often varies at different storage levels, typically corresponding to different seasons (Dang et al., 2020). Unlike the H06 approach, which employs a single release coefficient for a full year of

operation, both new algorithms consider a daily filling ratio, i.e. relative water storage ($Srel_d$), as defined by the following equation:

$$Srel_d = \frac{S_d}{C} \tag{6}$$

in which $S_d$ (km$^3$) is the reservoir storage on day 'd'.

In the scaling algorithm SA, daily release is computed as a function of daily relative storage and daily inflow into the reservoir. Daily inflow is scaled with the ratio of mean annual inflow ($\bar{I}$) to the 30-day mean inflow ($\bar{I}_{30d}$). This ratio is used to

represent the general effect of reservoirs to alter the temporal variation of streamflow by storing excess water during high-flow months and releasing it during low-flow months. Multiplication of $\bar{I}$ with $Srel_d$ mimics a prompt response to extreme events where storage can fill up within a few days. The release in the SA algorithm when water storage is at level $n$ is calculated as follows:

$$R_d = p_n \cdot \left[ Srel_{d-1} \cdot \bar{I} + \frac{\bar{I}}{\bar{I}_{30d}} \cdot I_d \right] \quad \text{for } n = 1, 2, 3 \tag{7}$$

in which $\bar{I}_{30d}$ (m$^3$/s) represents the mean inflow into the reservoir during the last 30 days. The variable $n$ indicates the storage

level at time $d$-$1$, and $p_n$ is the parameter assigned to storage level $n$ (one parameter assigned to each storage level). Levels 1, 2, and 3 correspond to $Srel$ as follows: Level 1 for above 0.7, Level 2 for between 0.4 and 0.7, and Level 3 for below 0.4. The paramters value need to be determined through the calibration process. These parameters enable us to adjust the mean release, while temporal variability is estimated inside the square brackets.

In contrast to the SA method, the weighting algorithm WA, does not consider $I_d$ but solely relies on $Srel_d$ for weighting

$\bar{I}$ and $\bar{I}_{30d}$ to compute the release. A maximum of 30% of $\bar{I}_{30d}$ contributes to release estimation at higher storage levels ($Srel \geq 0.7$), while it reaches 100% when the reservoir is empty, which is identical to run-of-the-river flow. In the WA algorithm when water storage is at level $n$, the release is estimated as follows:

$$R_d = q_n \cdot [Srel_{d-1} \cdot \bar{I} + (1 - Srel_{d-1}) \cdot \bar{I}_{30d}] \quad \text{for } n = 1, 2, 3 \tag{8}$$

where $q_n$ is the parameter assigned to storage level $n$ that needs to be determined. We opted for $\bar{I}_{30d}$ over $I_d$ assuming that release decisions may rather be based on the past inflow over a longer period and not on the inflow on just the previous day. Contrary to

the H06 approach, where the release is independent of inflow in reservoirs with large storage capacity relative to the annual inflow (meaning constant release throughout the year, see Eq. 5), both new algorithms consider the impact of inflow on release in all reservoirs. This impact varies with different seasons and storage levels, leading to variability in release throughout the year, which is more realistic (see Eq. 7 and Eq. 8). It should be noted that the new algorithms do not distinguish between irrigation





and non-irrigation reservoirs. This is because the estimation of downstream water demand at a large scale is generally very

uncertain, and reservoirs are usually designed for multiple purposes.

In each of the three algorithms, if $S_d$ falls below 10 percent of the storage capacity ($C$), the calculated $R_d$ is adjusted to $0.1 \cdot R_d$ if the available water is sufficient; otherwise, the entire $S_d$ will be released. Finally, the reservoir outflow is calculated as follows:

$$O_d = R_d + SP_d \qquad\qquad (9)$$

where $O_d$ (m$^3$/s) and $SP_d$ (m$^3$/s) are the reservoir outflow and the spill from the reservoir during day 'd', respectively. $SP_d$ is

calculated as the difference between $S_d$ and $C$, where $S_d$ exceeds $C$; otherwise, it is zero.

**2.3 Data**

The ResOpsUS dataset (Steyaert et al., 2022), which served for calibrating and evaluating the three algorithms in this study, encompasses daily records of inflow, storage, outflow, elevation, and evaporation for up to 679 US reservoirs. The available data spans from 1930 to 2020, determined by each dam's commissioning year and data availability. In this study, data on reservoir

inflow (daily), outflow (monthly), and storage (monthly) from 1980 to 2019 were considered, divided into two distinct periods: a calibration phase spanning from 1980 to 2009, and a validation phase covering the years 2010 to 2019. Monthly data were computed from daily records, excluding months with more than one week of missing values. Subsequently, we applied filters to the dataset, considering only reservoirs with a minimum data length of five years, a minimum reservoir capacity of 0.5 km$^3$, as well as ensuring there is only one reservoir per 0.5°×0.5° grid cell and no negative values. This resulted in 100 reservoirs, with

35 having data for storage, inflow and outflow and 65 having data for storage and outflow only. The minimum number of monthly data values for the 65 (35) reservoirs was 111 (252) for the calibration period and 65 (59) for the validation period. The reservoirs' storage capacities ($C$) range from 0.5 km$^3$ to 36.7 km$^3$ based on the GRanD dataset (Lehner et al., 2011). Out of the total 100 reservoirs, nine are irrigation reservoirs. Detailed information on each reservoir is provided in Table S1.

**2.4 Model variants and calibration approach**

The three reservoir operation algorithms were implemented in WaterGAP. For each algorithm, the algorithm-specific parameters were estimated by optimizing the Kling–Gupta Efficiency (KGE) (Kling et al., 2012), including the trend term (see Eq. 10). This optimization was performed through a single-objective calibration using the monthly time series of four variables: observed outflow, observed storage, observed storage anomaly, and estimated storage (based on observed storage changes and reservoir capacity, as detailed below). As in previous studies by Dong et al. (2023), Turner et al. (2021), and Shin et al. (2019), the

uncalibrated H06 (DH) is used as a benchmark. For comparison purposes, all calibrated algorithms used the inflow into reservoirs simulated by the DH algorithm to ensure that the same inflow data were applied across all algorithms. To achieve this, we first ran WaterGAP with the DH algorithm and saved the reservoir inflow data. Later, these inflow data were read from files and used as the inflow source to model each reservoir independently. As a result, when applying the CH, SA, and WA algorithms, the operations of upstream reservoirs did not affect downstream reservoirs. The calibration runs were initialized by running

WaterGAP five times for the year 1979 to allow water storages to reach a relatively stable equilibrium state.

In addition to the inflow simulated by WaterGAP, we also assessed the algorithms based on observed inflow where available. This was done to check the performance of reservoir operation algorithms in the presence of high-quality inflow data, as the performance of the algorithms may be heavily impacted by poor inflow data (Vanderkelen et al., 2022). Moreover, we assessed the impact of distinguishing irrigation and supply reservoirs from other reservoirs. The distinction for irrigation reservoirs is the





default approach for the H06 algorithm; however, here we also applied this distinction for supply reservoirs, as also their outflow depends on downstream demand. To this end, we modeled 21 reservoirs (nine irrigation and 12 supply reservoirs) in two different ways for all algorithms: one including downstream demand and the other without considering it. The purpose of this comparison is to evaluate whether including downstream demand, despite the high uncertainty in water demand estimation for the reservoirs, enhances the outflow and storage simulation, or whether it may not add value and instead introduce unnecessary complexity. In

the case of the SA and WA approaches for considering downstream demand, similar to the DH algorithm, Eq. 4 was used with the default value for the parameter $a_2$. However, instead of using $\overline{I'}$, $\overline{I}$ was applied in Eq. 4. The resulting $R'_d$ from Eq. 4 then replaced $\overline{I}$ in Eqs. 7 and 8 for estimating $R_d$.

Table 1 shows a summary of the different calibration variants. In Table 1, each calibration variant is characterized by a combination of a reservoir operation algorithm, a calibration variable, an inflow source, and whether or not downstream demand

is considered. For example, calibrating the CH algorithm against outflow using inflow simulated by WaterGAP while considering downstream water demand represents one calibration variant. Thus, each reservoir operation algorithm comprises 12 calibration variants (eight utilizing WaterGAP inflow and four using observed inflow), leading to a total of 36 calibration variants.

**Table 1.** Components of the different calibration variants, comprising 36 variants in total, with 12 variants for each algorithm. Each algorithm includes four variants using WaterGAP inflow with downstream demand considerations (calibrated against

outflow, storage, storage anomaly, and estimated storage), four variants using WaterGAP inflow without downstream demand, and four variants using observed inflow. Each calibration variant is defined by the combination of a reservoir operation algorithm, calibration variable, inflow source, and the consideration or non-consideration of downstream demand. For CH, the default approach incorporates the downstream demand of irrigation reservoirs, while the opposite is true for SA and WA. Additionally, considering the downstream demand for supply reservoirs is not the default approach for any of the operation algorithms. For

calibration variants that utilize observed inflow, only the default approach of each algorithm is considered.

| Operation algorithm | Calibration variable | Inflow source | Downstream demand considered? |
|---|---|---|---|
| CH | Outflow | WaterGAP | Yes[1] |
| | Storage | | No |
| | Storage anomaly | Observation | Yes[2] |
| | Estimated storage | | |
| SA | Outflow | WaterGAP | Yes[1] |
| WA | Storage | | No |
| | Storage anomaly | Observation | No |
| | Estimated storage | | |

[1] Water demand is considered for irrigation and supply reservoirs, i.e., 21 out of 100 studied reservoirs.
[2] Water demand is considered for irrigation reservoirs, i.e., two out of 35 studied reservoirs with observed inflow.

The parameters of each algorithm were calibrated using a grid search approach, with the parameter range detailed in Table S2. The parameter estimation using storage anomalies and estimated storage serves as the main experiment, as the primary

emphasis of this study is on exploring the added value of incorporating storage anomalies or changes into the calibration of reservoir operation algorithms. For the storage anomaly, the mean storage during the calibration period is subtracted from the data. However, the storage anomaly lacks information about the bias term, potentially resulting in a time series that considerably deviates from the actual absolute water storage. Having actual absolute storage is advantageous, as reservoirs are the only surface water bodies for which we can model absolute storage within the WaterGAP. To provide an alternative, we calculated the

"estimated storage time series", which refers to storage values that are not observed directly but are estimated using observed storage changes and the reservoir capacity *C*. First, we determined the storage changes time series by subtracting the initial month's storage value from the observed monthly storage values. Assuming the reservoir reaches its maximum capacity at least once between 1980 and 2009, we calculated the maximum observed monthly storage change, termed *Difmax*. We then subtracted





*Difmax* from the GRanD reservoir storage capacity to estimate the initial water storage for the first month. The estimated storage

time series is then obtained by adding the storage changes to this estimated initial storage. Since the data are on a monthly scale and maximum storage on a daily scale are generally greater than those on a monthly scale, we applied a scaling factor of 1.2 to *Difmax*. This adjustment means that *Difmax* used in our calculations is 20% higher than the initially calculated value. The 20% scaling factor is derived from the mean difference between the maximum daily storage and the monthly storage observed in 100 studied reservoirs (see Table S1).

**2.5 Performance evaluation metrics**

The performance of the reservoir operation algorithms was evaluated using KGE and the normalized root mean square error (nRMSE). KGE is widely used for model calibration and evaluation, as it simultaneously considers multiple important aspects of model performance, providing a comprehensive assessment (Beck et al., 2019; Lamontagne et al., 2020). The use of nRMSE offers additional insights by focusing on the magnitude of errors. Following Hosseini-Moghari et al. (2020), we incorporated the

trend component into the conventional KGE equation as follows:

$$KGE = 1 - \sqrt{(R_{KGE} - 1)^2 + (B_{KGE} - 1)^2 + (V_{KGE} - 1)^2 + (T_{KGE} - 1)^2} \tag{10}$$

$$R_{KGE} = \frac{cov(sim, obs)}{\sigma_{sim} \cdot \sigma_{obs}} \tag{11}$$

$$B_{KGE} = \frac{\overline{sim}}{\overline{obs}} \tag{12}$$

$$V_{KGE} = \frac{\sigma_{sim}/\overline{sim}}{\sigma_{obs}/\overline{obs}} \tag{13}$$

$$T_{KGE} = \frac{T_{sim}}{T_{obs}} \tag{14}$$

where $R_{KGE}$ represents the correlation coefficient between observed (*obs*) and simulated (*sim*) time series; $B_{KGE}$ denotes the bias of the mean simulated ($\overline{sim}$) compared to the mean of observed ($\overline{obs}$), $V_{KGE}$ is the variability component that denotes the ratio of the standard deviation of the simulated ($\sigma_{sim}$) to the standard deviation of the observed ($\sigma_{obs}$) time series, divided by their mean, and $T_{KGE}$ represents the ratio of the linear trend of the simulated time series ($T_{sim}$) to the observed one ($T_{obs}$). In the case

of calibrating against storage anomaly, we did not divide $\sigma$ by the mean, as the mean for storage anomalies is zero. Similarly, the $B_{KGE}$ component was not considered in calculating KGE related to storage anomalies. The optimal value for the KGE and its four components is 1. The KGE range is $(-\infty, 1]$, while $R_{KGE}$ ranges from -1 to 1; $B_{KGE}$, $V_{KGE}$ and $T_{KGE}$ can vary between $-\infty$ and $+\infty$. Following Knoben et al. (2019), a KGE value above -0.73 indicates that the model performs better than the mean of observations if the trend component is included in the KGE.

The normalized root mean square error (nRMSE) is calculated as:

$$nRMSE = \frac{\sqrt{\frac{1}{T}\sum_{t=1}^{T}(obs_t - sim_t)^2}}{\sigma_{obs}} \tag{15}$$

The perfect value for nRMSE is zero. Normalizing the RMSE with the standard deviation of observations brings this metric closer to the Nash-Sutcliffe Efficiency (NSE), but different from the NSE, the nRMSE cannot become negative (Turner et al., 2021).



## 3 Results

### 3.1 Performance of calibration variants in the case of simulated inflow into reservoirs

We found that calibrating against observed water storage, water storage anomalies, or estimated water storage (derived from observed water storage changes and GRanD storage capacity) improves the very poor simulation of storage by the calibration-free algorithm (DH) for both the calibration and validation periods in the case of all three algorithms (Table 2). In the case of DH, storage simulation is skillful, i.e. with a $KGE_{storage} > -0.73$, for only 16% of the 100 reservoirs during the calibration period, and for 15% during the validation period. Calibration of the H06 reservoir operation algorithm (CH) achieves skillful storage simulations for 64% (39%) of the reservoirs when calibrated against storage anomalies and for 69% (32%) of the reservoirs when calibrated against estimated storage during the calibration (validation) period. Both SA and WA outperform CH in storage simulation when calibrated against storage-related variables for both the calibration and validation period (Table 2 and Fig. 1). However, the fit of simulated to observed storage remains poor during the validation period, in particular after calibration against storage anomalies and estimated storage (Table 2 and Fig. 1).

**Table 2.** The number of reservoirs out of 100 in which KGE values are greater than the benchmark thresholds of -0.73 during the calibration (validation) phase. All algorithms were calibrated against outflow, storage, storage anomaly, as well as estimated storage using KGE as the objective function. The inflow data is sourced from the WaterGAP model.

| Calibrated variable | Algorithm | KGE > -0.73 | |
| --- | --- | --- | --- |
| | | Outflow | Storage |
| — | DH | 63 (56) | 16 (15) |
| Outflow | CH | 78 (68) | 22 (30) |
| | SA | 86 (71) | 14 (24) |
| | WA | 86 (69) | 20 (30) |
| Storage | CH | 68 (69) | 91 (46) |
| | SA | 66 (67) | 98 (68) |
| | WA | 67 (66) | 100 (55) |
| Storage anomaly | CH | 67 (69) | 64 (39) |
| | SA | 67 (69) | 68 (45) |
| | WA | 71 (70) | 66 (45) |
| Estimated storage | CH | 70 (69) | 69 (32) |
| | SA | 65 (68) | 69 (46) |
| | WA | 67 (70) | 74 (41) |

Calibration against storage-related variables only slightly improves the mostly poor simulations of reservoir outflow during the calibration period and shows a bit more improvement in the validation period (Table 2 and Fig. 1). Skillful outflow simulations were achieved for 86% of the reservoirs when either SA or WA were calibrated against outflow, compared to 78% for CH and 63% for DH during the calibration phase. However, skillful storage simulations were observed in only 14% (24%) and 20% (30%) of the reservoirs for SA and WA, respectively, compared to 22% (30%) for CH and 16% (15%) for DH in the calibration (validation) phase (Table 2). The performances of outflow simulations with CH, SA and WA are very similar in both the calibration and validation periods except in the case of calibration against observed outflow for the calibration period. In this case, SA and WA achieve positive positive $KGE_{outflow}$, with medians of 0.15 for SA and of 0.13 for WA. Calibrating against outflow improves the correlation, variability and trend of the simulated outflow compared to DH for all three algorithms, while the bias is not affected much (Figs. S1-S4). On average, outflow trends are underestimated. Calibrating against outflow weakens the correlation, variability and trend of the storage simulation for all three algorithms (except the trend in the case of CH), while strongly improving the bias as compared to DH (Figs. S1-S4), so that model performance regarding storage is not affected in a





relevant manner by calibration against outflow and remains very poor. All these results suggest that when algorithms are

calibrated against outflow, the mean observed storage generally remains a better estimator than the simulated storage.

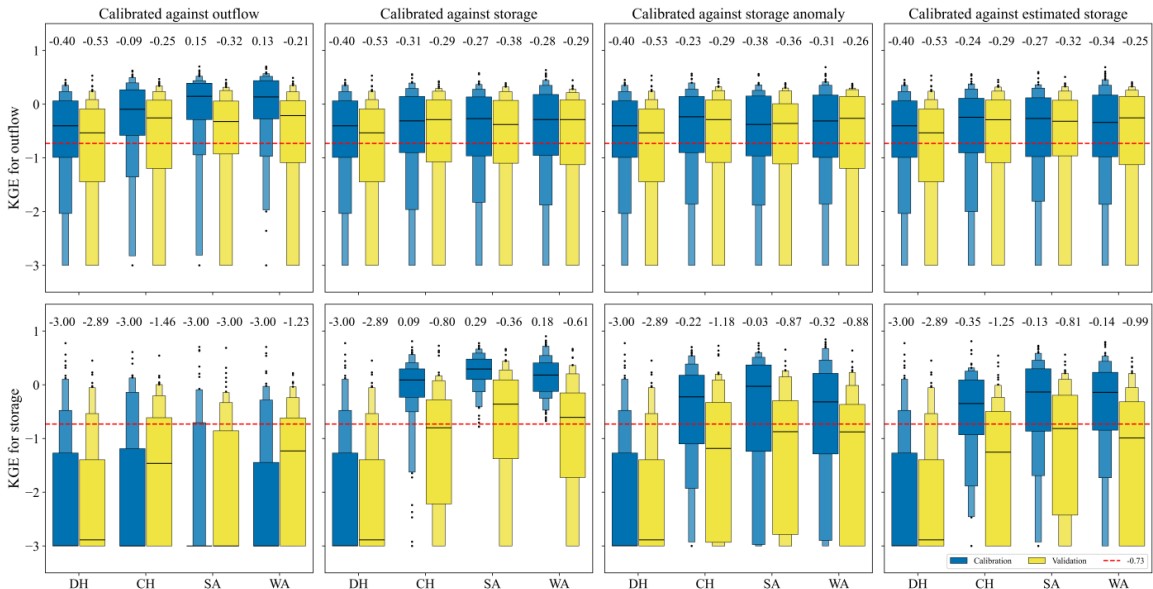

**Figure 1.** Letter-value plots of KGE for outflow and storage of 100 studied reservoirs for DH, CH, SA, and WA algorithms for
the calibration period (1980-2009, in blue) and validation period (2010-2019, in yellow). All algorithms are calibrated against
outflow (first column), storage (second column), storage anomaly (third column), as well as estimated storage (fourth column)
using KGE as the objective function. The values at the top of the panels are the median KGE (indicated by the horizontal line).
The dashed red lines represent the KGE benchmark threshold of -0.73. KGE values less than -3 are set to -3. The widest box
contains 50% of the 100 data points, the second widest 25% of the data (12.5% in the upper box and 12.5% in the lower box),
the third widest 12.5%, and so on. The inflow data is sourced from the WaterGAP model.

Calibrating against storage (second column of Fig. 1) leads to the highest $KGE_{storage}$ values; with a median $KGE_{storage}$ of 0.29,

SA outperforms CH and WA, while the $KGE_{outflow}$ and its component values for the three algorithms are similar (Figs. S1-S4).

Calibrating against storage anomaly (third column in Fig. 1) or estimated storage (fourth column in Fig. 1) improves both storage

and outflow simulations as compared to DH but the fit to observed storage is considerably worse than in the case of calibration

against storage. While the median $KGE_{storage}$ in the case of calibration against storage anomaly is slightly better than when

calibrated against estimated storage, the widest box of the letter-value plot related to calibration against estimated storage, which

contains 50% of the data, is above the one for calibration against storage anomaly. The improvement of storage simulation is

mainly through bias adjustment (Fig. S2). The DH algorithm has a median $B_{KGE}$ of 1.90 for storage during the calibration period.

This value decreases to 0.92 (1.04, 0.99), 0.71 (0.91, 1.18), and 1.25 (1.44, 1.32) for calibration against storage, storage anomaly,

and estimated storage of the CH (SA, WA) algorithm, respectively. The correlation is improved in the case of SA and WA but

only in the calibration period (Fig. S1). The variability is improved for calibration against storage anomaly, while calibration

against estimated storage leads to an underestimation of storage variability (Fig. S3). By calibration against storage, storage

anomaly and estimated storage, the trend component of $KGE_{storage}$ strongly improves as compared to DH for the calibration period

but the trend is on average still underestimated (Fig. S4). Assessing the $KGE_{storage\_anomaly}$ when calibrating with different variables

shows less degradation during the validation phase (Fig. S5). For example, the number of skillful simulations for storage reached





17 (18), 93 (44), 98 (59), and 99 (55) when calibrating using storage anomaly with DH, CH, SA, and WA, respectively (see Table 2 for comparison).

The fit to observed storage-related variables is much less improved as compared to DH for the validation period than for the calibration period (Table 2 and Fig. 1). Comparing calibration against storage anomaly and estimated storage, which are the available options when using only remote sensing data, reveals that SA and WA are preferable to CH and DH, even though the

differences from CH are small during the validation period. Differences between the $KGE_{storage}$ values of SA and WA are small for all calibration variables for both calibration and validation periods.

Examining the empirical cumulative distribution functions (eCDFs) for nRMSE reveals that the eCDFs for outflow are much closer across different algorithms compared to those for storage (see Fig. 2). This suggests that calibration has a more significant impact on storage than on outflow. Calibration against any storage-related variable generally enhances outflow performance at

lower $nRMSE_{outflow}$ levels (in approximately 60% of the reservoirs), while at higher $nRMSE_{outflow}$ ranges, a slight degradation is observed in about 35% of reservoirs (with probabilities ranging from less than 0.60 to 0.95, mainly concentrated between 0.8 and 0.9). When calibrating against outflow, there is generally improvement in $nRMSE_{storage}$ for CH and WA algorithms, while no clear improvement is seen for SA. Moreover, the error in outflow simulation is reduced in over 40% of reservoirs where the $nRMSE_{outflow}$ was already lower compared to others. For $nRMSE_{outflow}$ greater than 0.98, there is almost no discernible

improvement observed when calibrating algorithms against outflow, as indicated by the eCDFs. The calibration against storage anomaly, which is the main calibration variant, especially in the validation phase, reveals that SA performs better than WA. SA shows $nRMSE_{storage}$ lower and nearly similar $nRMSE_{outflow}$ compared to WA. Disregarding the magnitude of error, the eCDF for validation has a shape similar to that of the calibration period, suggesting that the error distribution for the algorithm is consistent across both periods.

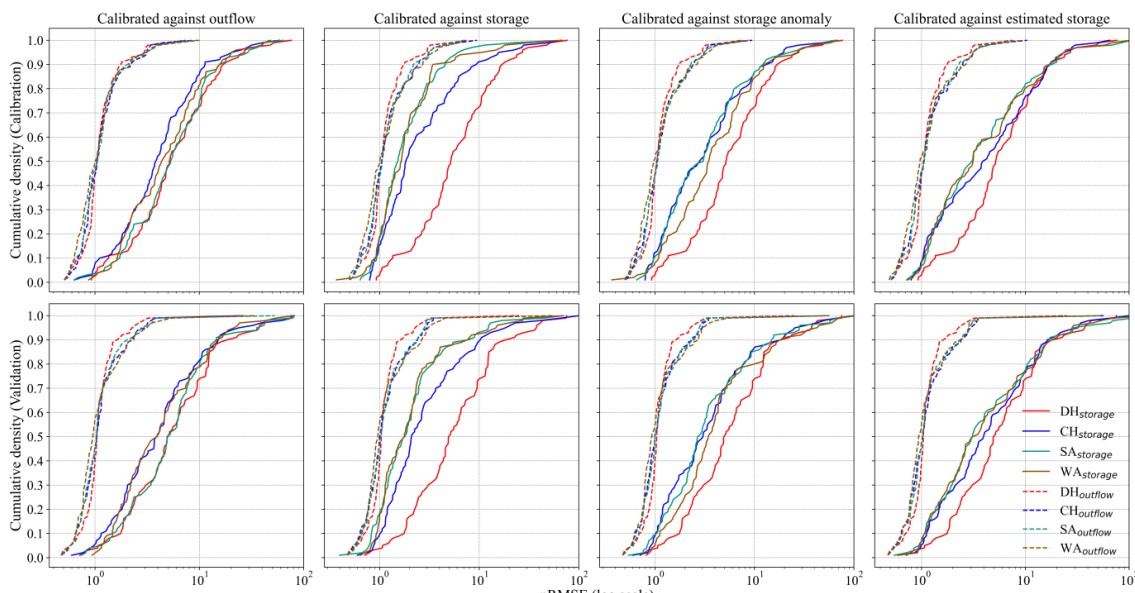


**Figure 2.** Empirical cumulative distribution functions of nRMSE for storage and outflow of 100 studied reservoirs are based on DH, CH, SA, and WA algorithms for the calibration period (1980-2009) and validation period (2010-2019). All algorithms are calibrated against outflow (first column), storage (second column), storage anomaly (third column), and estimated storage (fourth column) using KGE as the objective function. The x-axis has a logarithmic scale. If nRMSE is larger than 1, the mean error is larger than the standard deviation of the observational values. The inflow data is sourced from the WaterGAP model.






### 3.2 Illustrative calibration results for three reservoirs

As an example, we plotted the time series of storage and outflow for the Glen Canyon Dam (Lake Powell) in Figure 3. This dam is one of the largest in our study, with several dams located upstream. The WaterGAP dataset includes four upstream reservoirs as global reservoirs, with storage capacities ranging from 0.57 to 4.3 km³. Calibrating the H06 model against outflow did not

lead to better results compared to the DH model (Figs. 3a, 3b). However, some improvement was observed in the outflow simulation for SA and WA during the calibration period, though this led to worse outflow simulation during the validation phase. Despite this, with a KGE > -0.73, all outflow simulations demonstrated skillful performance. Calibration against outflow did not degrade storage simulation compared to the DH, except for SA, particularly during the validation phase, where the variability of the simulated time series was more than three times higher than the observed one (Table S3). During the calibration phase,

storage levels are mainly above 40% (10 km³) of the capacity, with a sharp decline between 40% and 70%, and smaller changes when the reservoir is filled above 70% (17.5 km³). This pattern leads to storage levels below 40% not being adequately considered in the parameter selection process. As a result, when storage drops below 10 km³ during the validation phase, the outcomes are not promising. The large difference between the capacity reported by GRanD (25 km³) and the maximum observed daily storage (31.7 km³) results in poorer performance in storage simulation for all calibrated algorithms based on estimated storage compared

to storage anomaly. This ~20% difference between the reported capacity and maximum observed storage introduces a 20% bias, which directly impacts the bias and variability components of $KGE_{storage}$ (Table S3). However, there is almost no bias in the outflow, thanks to the data from the Lees Ferry station, located just downstream of the dam, which is used in the bias adjustment of streamflow simulations in WaterGAP through a simple calibration approach.

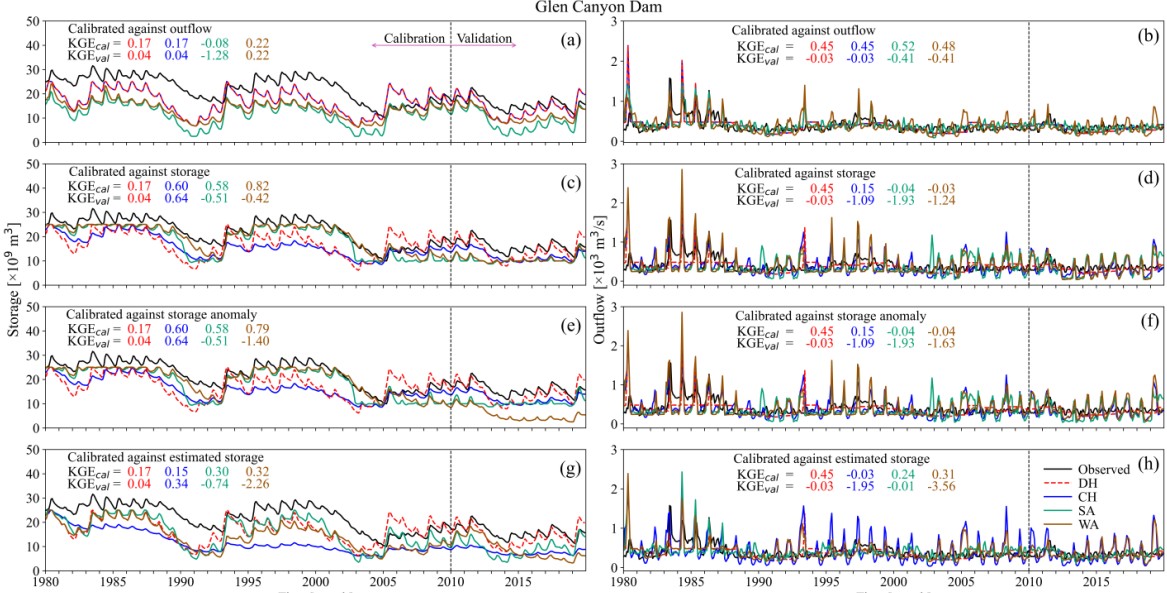

**Figure 3.** Monthly time series of observed storage and outflow, as well as simulated values from DH, CH, SA, and WA algorithms for Glen Canyon dam, GranD ID 597, calibrated against (a, b) outflow, (c, d) storage, (e, f) storage anomaly, and (g, h) estimated storage using KGE as the objective function. The dashed black lines distinguish between the calibration and validation periods. The inflow data is sourced from the WaterGAP model.



Very poor storage simulation with a much higher seasonal magnitude compared to observed storage is seen for the Yellowtail

Dam (GranD ID = 355), an irrigation reservoir with different calculations in the DH and CH algorithms compared to the SA and

WA algorithms, and for the Harry S. Truman Dam (GranD ID = 989), which is a hydropower reservoir (Fig. 3). Calibrating

against storage anomalies can lead to time series of storage with considerable bias (Fig. 3e). This issue can also occur when

calibrating against estimated storage if there is an offset between the estimated storage and the in-situ observation (Fig. 2g). The

time series related to the Yellowtail Reservoir reveals that SA and WA, which do not consider the irrigation purpose of this

reservoir, can simulate reservoir storage better than DH and CH which explicitly take into account the downstream water demand

(Fig. 3a). However, the opposite is true for outflow simulation, where the uncalibrated DH performs best (Fig. 3b).

From these examples, we found that calibrating solely against storage-related variables does not necessarily lead to poorer

outflow simulations (Fig. 4e–h). However, other factors, such as inaccuracies in reservoir capacity data (e.g., for the Glen Canyon

Dam) and discrepancies between actual available water and the reported static storage value in the GranD dataset — which may

include dead storage (see Table S1 for Yellowtail and Harry S. Truman dams) — are important considerations when evaluating

the performance of the reservoir operation algorithm. In such cases, comparing storage anomalies may offer a more reasonable

assessment than comparing absolute storage. This error in storage simulation may also affect outflow simulations, where

inaccuracies in input data are the primary factor leading to inaccurate storage levels being maintained during the validation phase

(Fig. 3e).

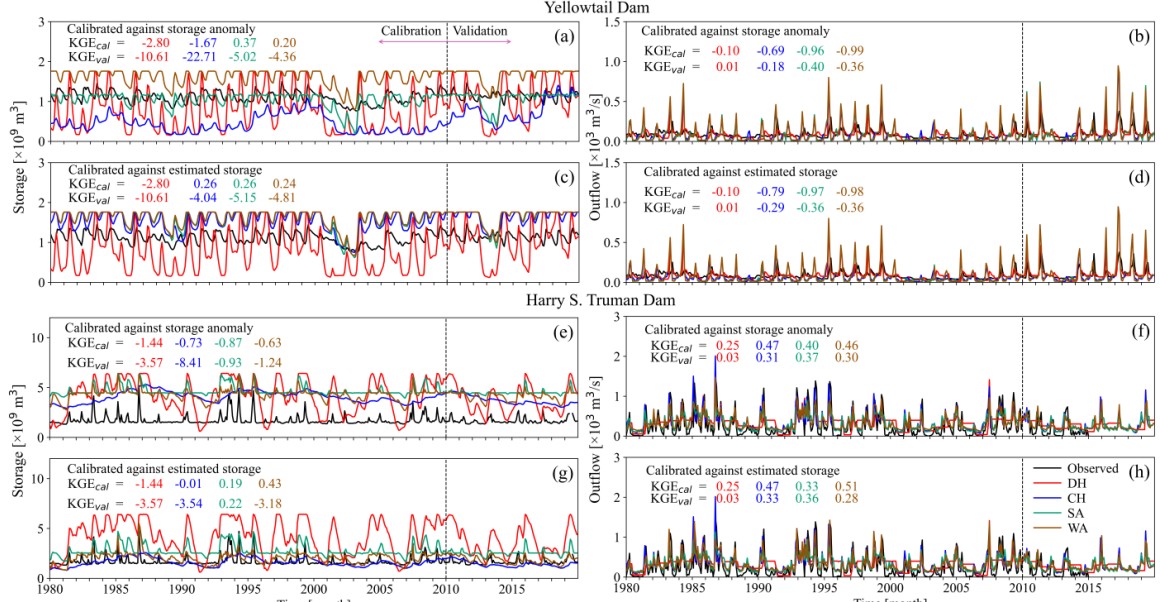


**Figure 4.** Monthly time series of observed storage and outflow, as well as simulated values from DH, CH, SA, and WA
algorithms for Yellowtail/Harry S. Truman reservoirs, GranD IDs 355/989, calibrated against (a, b)/(e, f) storage anomaly and
(c, d)/(g, h) estimated storage using KGE as the objective function. The primary purposes of the Yellowtail Dam and the Harry
S. Truman Dam are irrigation and hydropower, respectively. The dashed black lines distinguish between the calibration and
validation periods. The inflow data is sourced from the WaterGAP model.



**3.3 Impact of using observed streamflow as input to the reservoir operation algorithms**

Comparing the results of the modeling using WaterGAP inflow and observed inflow is presented in Fig. 5 for 35 reservoirs of 100 studied ones. Based on Fig. 5, there is no overall improvement or deterioration in storage simulation when using observed or WaterGAP inflow data, except for the WA algorithm, which demonstrates better performance with observed inflow than with simulated streamflow. This is evident as most of the circles are positioned above the y=x line (Fig. 5c). However, performance of WA with observed inflow is not better than the performance of SA. In contrast, there is a considerable improvement in the reservoir outflow simulation when utilizing observed inflow data. For instance, KGE$_{outflow}$ below -1 achieved with WaterGAP inflow can approach 1 with observed inflow (Fig. 5f). In most cases, KGE$_{outflow}$ between 0-0.5 based on WaterGAP inflow reaches 0.5-1 based on observed inflow. The most substantial improvement is observed for the WA algorithm, where the median of KGE$_{outflow}$ across various calibration objectives, ranging from [-0.27, 0.14], increases to [0.56, 0.69] upon replacing WaterGAP inflow with observed data. It implies that the WA is more sensitive to the quality of inflow data than other algorithms. The same pattern is reiterated during the validation period, with the median KGE$_{outflow}$ [0.38, 0.56] compared to [-0.87, -0.41] based on observed inflow compared to WaterGAP inflow across all calibration variants (Fig. S7). Using observed inflow improves almost all components of KGE$_{outflow}$, but the main components that are improved are variability and trend components (see Figs. S8-S15).

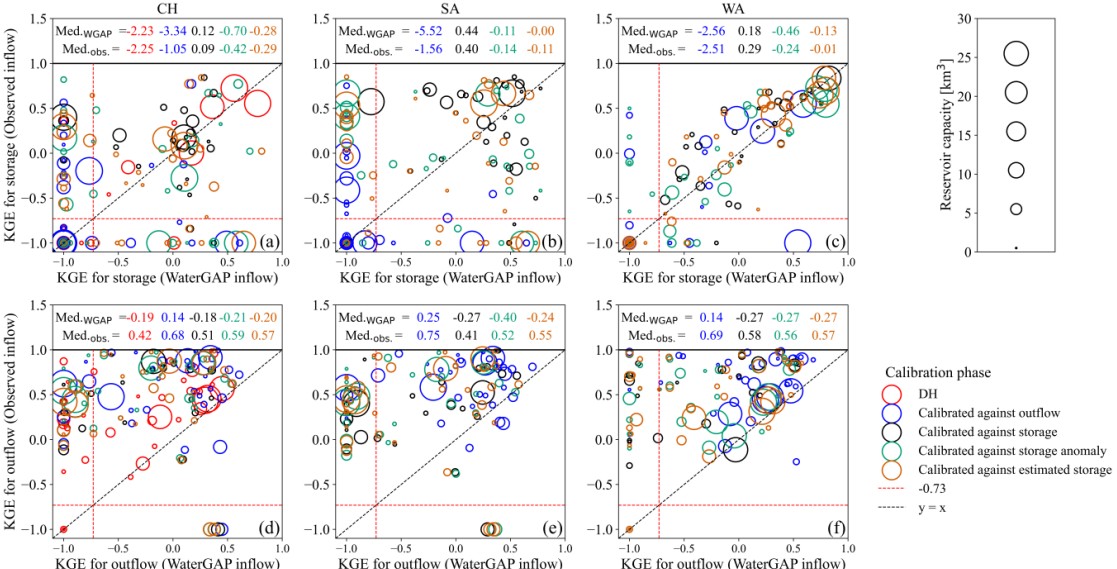

**Figure 5.** The relationship between KGE of (a-c) storage and (d-f) outflow obtained from modeling reservoirs using WaterGAP inflow and observed inflow to the reservoirs for the calibration period (1980-2009) for 35 reservoirs with observed inflow. KGE values less than -1 are set to -1. The KGE values for storage anomaly and estimated storage are not shown. The circle size indicates the reservoir capacity. The values above each panel indicate the median KGE, with the top values achieved with WaterGAP inflow and the bottom values with observed inflow. The dashed red lines indicate the KGE benchmark threshold of -0.73.

**3.4 Impact of considering downstream water demand**

We evaluated the benefit of distinguishing irrigation and water supply reservoirs from others by counting how many times estimating the outflow of irrigation reservoirs (9 reservoirs) and supply reservoirs (12 reservoirs) using Eq. 4 (the default





approach for irrigation reservoirs in the H06 algorithm, which takes into account the seasonality of downstream water demand)

leads to a more skillful simulation compared to disregarding water demand in the modeling of reservoir dynamics. We found that

there is no general advantage in distinguishing irrigation and supply reservoirs from other reservoirs, in particular when

calibrating against storage anomalies or estimated storage using the overall superior WA and SA algorithms. While in the case

of calibration against estimated storage, the SA algorithm performs better for outflow with considering downstream demand, the

opposite is true for storage. For the WA algorithm, the same number of reservoirs achieve better or worse streamflow performance

if taking into account downstream water demand while storage performance is better if demand is not considered (Table 3).

**Table 3.** The number of irrigation and supply reservoirs (out of 21) where KGE values for the calibration phase are higher when considering downstream water demand than when neglecting downstream demand. Improvements are only identified if the achieved KGE value is larger than -0.73, i.e. the simulation is skillful. The values in parentheses indicate the number of reservoirs

where neglecting downstream demand leads to higher KGE values. All algorithms are calibrated against outflow, storage, storage anomaly, and estimated storage using KGE as the objective function. The inflow data is sourced from the WaterGAP model.

| Calibrated variable | CH | | SA | | WA | |
|---|---|---|---|---|---|---|
| | Outflow | Storage | Outflow | Storage | Outflow | Storage |
| Outflow | 4 (5) | 1 (3) | 10 (6) | 1 (3) | 8 (8) | 3 (4) |
| Storage | 7 (6) | 7 (2) | 5 (6) | 11 (10) | 6 (6) | 7 (14) |
| Storage anomaly | 6 (6) | 3 (3) | 7 (6) | 7 (10) | 8 (6) | 7 (9) |
| Estimated storage | 6 (4) | 6 (1) | 9 (3) | 6 (10) | 6 (6) | 3 (12) |

## 4 Discussion

### 4.1 Value of calibration and choice of reservoir operation algorithm

Applying streamflow simulated by the global hydrological model WaterGAP 2.2e as inflow to 100 US reservoirs, we found that

the outflow generated by the calibration-free algorithm DH is a better alternative to the mean observed outflow. However, the

opposite is true for simulated reservoir storage (see Fig. 1), underscoring the need for reservoir-specific calibration. Our findings

indicate that all three calibrated algorithms generally perform better than DH in terms of storage, but the effect on reservoir

outflow simulation is negligible. The degree of improvement varies considerably between reservoirs, and in some cases, no

improvements are seen, as also reported by Turner et al. (2021) with a more complex reservoir operation algorithm. Among the

calibrated algorithms, SA and WA performs better than CH when calibrated against storage, storage anomaly, and estimated

storage. Thus, CH may only be preferred over SA and WA in the case of irrigation reservoirs with rather good water demand

information or if computational resources are very limited as CH requires the estimation of only two instead of three parameters

for non-irrigation reservoirs. While the performance of SA and WA cannot be distinguished by KGE, nRMSE indicates a slightly

better performance of SA in the case of calibration against storage anomalies (Fig. 2).

480       Calibration of H06 reveals that default parameters are rarely included in the calibrated parameter sets (Fig. S16), especially

noticeable for irrigation reservoirs where parameter $a_2$ almost always remains at its lower bound of 0.1. According to Eq. 4, this

implies that calibration prioritizes using a scaled version of long-term inflow rather than directly integrating demand through

addition. The demand estimation is not accurate enough for reservoir operations, resulting in increased complexity with limited

benefit when distinguishing irrigation and supply reservoirs from other types of reservoirs (Table 3). Vanderkelen et al. (2022)

similarly observed minimal additional value in including irrigation demand in reservoir operations.





### 4.2 Calibration variables

Calibrating against outflow not only fails to improve storage simulations but also leads to a degradation in SA performance in storage simulation during calibration phase. In contrast, calibrating against all types of storage-related variables slightly improves outflow compared to the DH algorithm (see Fig. 1 and Table 2). Thus, calibrating against storage-related variables is more effective than calibrating against outflow when aiming to improve the simulation of both variables through a single-objective calibration. Additionally, comparing the KGE values of the compromise solution (defined as the solution with the minimum Euclidean distance from the optimal KGE value of 1 for both storage and outflow) with KGE values from calibrations against storage and outflow indicates that the results of calibration against storage are considerably closer to the compromise solution compared to those for outflow (see Fig. S6). A similar pattern is observed for calibrations against both storage anomalies and estimated storage. This suggests that calibrating solely against storage-related variables yields results closer to the compromise solution than calibrating against outflow alone. One reason for this is the lower sensitivity of outflow simulations to calibration compared to storage simulations. This finding is encouraging because, unlike outflow data, storage anomalies can be estimated using remotely sensed data. The data length should exceed five years to be used effectively for this purpose (Otta et al., 2023). Although our results indicate that, in general, calibrating against storage anomalies improves the simulation of storage, using the absolute simulated storage from such calibrations should be done carefully, as these calibrations do not always guarantee an improvement in absolute storage.

Calibrating against estimated storage does not outperform calibrating against storage anomalies (see Fig. 1 and Table 2), although theoretically, it should provide results closer to calibration against storage. The reason for this, besides the inherent error in storage estimation, can be traced to discrepancies between the capacity information from GRanD and the maximum daily observed storage (median difference equals to ~25%). The maximum observed storage should be less than or equal to capacity unless during an overtopping period. However, comparing maximum daily storage data from ResOpsUS with reservoir capacity from GRanD shows notable differences in several cases (see Table S1). GRanD does not account for overtopping and may include inaccurate data (Steyaert and Condon 2024). Inconsistencies are also reported for reservoir area; Dong et al. (2023) reported that the actual reservoir polygons of Ertan Reservoir and Jinping I Reservoir are 69% and 50% larger than the GRanD polygons. Therefore, for those reservoirs, modeling reservoir operation using GRanD information should not lead to good results, particularly for absolute storage simulation. Consequently, absolute storage comparison may not be a fair approach for model performance assessment, although it remains valid for comparing different algorithms. An assessment of the degradation in KGE values obtained from calibration against estimated storage compared to calibration against actual storage reveals that the results from estimated storage closely match those from actual storage when the difference between the reservoir capacity reported by GRanD and the maximum daily observed storage is minimal. However, as this difference increases, the discrepancy between the results of the two calibration variants also grows (Fig. S17). It is important to note that calibration against storage anomalies does not exhibit a direct relationship with these differences in storages.

To the best of our knowledge, two global datasets — the Global Reservoir Storage (GRS) introduced by Li et al. (2023) and the GloLakes dataset by Hou et al. (2024) — currently provide monthly time series of estimated absolute storage using remotely sensed information, along with either a geostatistical model or a volume-elevation/area-volume relationship. We assessed the quality of their estimates for the absolute storage of the studied reservoirs. GRS covers all 100 studied reservoirs, while GloLakes includes only 57 of those 100 reservoirs. The median $KGE_{storage}$ (without the trend component) was 0.26 for GRS and 0.14 for GloLakes, indicating that neither dataset provides estimates accurate enough to be considered reliable for calibrating reservoir





operation algorithms against their estimated absolute storage (see Table S4). The $B_{KGE}$ components for GRS, with a median of 0.84, range from significant underestimation — such as for Norfork Dam (GranD ID 1042), where the mean estimated storage is only 2% of the observed value — to substantial overestimation, such as for Albeni Falls Dam (GranD ID 305), where the mean estimated storage is 45 times greater than the observed value. GloLakes, with a median $B_{KGE}$ of 1.49, performs slightly better in terms of extreme bias; the largest underestimation occurs at Santa Rosa Dam (GranD ID 1086), where the mean estimated storage

is only 35% of the observed value. Maximum overestimation for GloLakes is observed at the same dam (Albeni Falls Dam) but is less extreme compared to GRS, though still substantial. The $R_{KGE}$ and $V_{KGE}$ components of KGE for storage are better than $B_{KGE}$ in terms of extreme values. However, with medians of 0.63 and 0.84 for GRS and 0.71 and 0.47 for GloLakes, respectively, $R_{KGE}$ and $V_{KGE}$ for both datasets are still not sufficiently promising, indicating uncertainty in remotely sensed storage anomaly estimates.

### 535    4.3 Relevance of the quality of simulated reservoir inflow and reservoir storage capacity data

We found that the quality of inflow data is more important than the reservoir operation algorithms for outflow simulation, while it has less impact on storage simulation. This finding aligns with Vanderkelen et al. (2022), who attributed the similar performance of natural lake parameterization and H06 to poor simulated streamflow in the Community Land Model. Using observed inflow as a substitute for simulated outflow (ignoring the dam) and comparing it with observed outflow reveals that the

DH algorithm, with median $KGE_{outflow}$ values of 0.42 (calibration) and 0.02 (validation), results in worse outflow simulations compared to the observed inflow, which has median $KGE_{outflow}$ values of 0.57 (calibration) and 0.36 (validation). This is in line with Vora et al. (2024), who reported that ignoring reservoirs in modeling may lead to better outflow simulations than DH in some cases. However, some skill is observed in other algorithms, particularly SA, where the median $KGE_{outflow}$ values for CH, SA, and WA are 0.68 (0.46), 0.75 (0.52), and 0.69 (0.56) for calibration (validation), respectively, when calibrated against

outflow (see Figs. 5 and S7). In contrast to Vanderkelen et al. (2022), our study found that using observed inflow did not lead to a clear improvement in storage simulation. One possible reason is the error in GRanD data, with a median difference of ~14% between GRanD data and maximum daily observed storage for reservoirs with observed data. Another potential reason could be the impact of initial storage on simulation outcomes, which varies depending on the level of regulatory of reservoir operations, as reported by Yassin et al. (2019).

### 550    4.4 Complexities of reservoir operations and dynamics

In addition to poor inflow data and inaccurate capacity information, other factors also impact the performance of reservoir modeling algorithms. Incorporating human decision-making into the model is very challenging, despite its critical importance (Rougé et al., 2021). This complexity arises because human decisions do not always follow operational rules due to evolving conditions, such as changes in water demand (Shah et al., 2019) or during droughts and floods (Nazemi and Wheater, 2015). For

example, the Hoover Dam (Lake Mead) and Glen Canyon Dam (Lake Powell) are interconnected, and historically, Glen Canyon could release enough water to meet downstream needs until 2014. However, due to a drought in 2012 and 2013, the release from Glen Canyon Dam in 2014 dropped to its lowest level since the initial filling of Lake Powell in 1963 (Arizona Water Resource, 2013; Colorado River Drought, 2019). This reduction in release was aimed at recovering Lake Powell's storage, which had fallen to ~40% of its capacity (NASA Earth Observatory, 2014). Additionally, climate change and increases in water demand can lead

to non-stationary situations, meaning that calibrated algorithms may not perform as well compared to the calibration period. This trend is observed in the ResOpsUS dataset, where there is generally a decreasing trend in reservoir storage, which also impacts





release (Steyaert and Condon, 2024). For example, the Hoover Dam has experienced a continuous negative trend in its capacity since 2000 (see Fig. S18). Understanding these trends is crucial for assessing the degradation of the studied algorithms during the validation period, where the connection between observed inflow and outflow also becomes weaker.

**4.5 Limitations**

In this study, we modeled each reservoir independently, which may affect the quality of the analysis. In practice, a calibrated upstream reservoir would lead to different inflows to a downstream reservoir. However, since the calibration has not had a considerable impact on outflow simulation, it is expected that the overall conclusions would be similar. For the SA and WA algorithms, a reservoir may reach relative storage level(s) (see Eqs. 7 and 8) during the validation phase that were not observed
during the entire calibration period. Consequently, the parameters for these unseen relative storage levels cannot be determined and are set to the lowest value (0.1 for both SA and WA). As a result, the performance of the algorithm for those reservoirs during the validation phase is affected by setting these undetermined parameters to the lowest value. In the case of the SA algorithm, this issue occurs for at most four reservoirs across the calibration variants, while for the WA algorithm it occurs for up to nine reservoirs (see Table S5). Moreover, although Yassin et al. (2019) suggest that a five-year spin-up period is generally
sufficient to fully stabilize even for large dams, and we used five simulations of 1979 as our spin-up period, a longer run extending further back before 1980 could result in different initial storage conditions. Consequently, this could affect the performance of the operational algorithm. This potential limitation should be acknowledged, as it may impact the accuracy and generalizability of the results.

**5   Conclusions**

In this study, we compared four reservoir operation algorithms: the widely used Hanasaki algorithm with default parameter values (DH) and three individually calibrated algorithms (the calibrated Hanasaki algorithm (CH), the scaling algorithm (SA), and the weighting algorithm (WA)). We use observations from the in-situ ResOpsUS dataset of 100 reservoirs across the USA as a benchmark. The goal was to determine how calibrating reservoir operation algorithms against observations can best improve the simulation of reservoir dynamics in continental and global hydrological models. All algorithms were implemented in the
global hydrological model WaterGAP. While CH is the calibrated version of DH and includes two calibration parameters for non-irrigation reservoirs and three for irrigation reservoirs, the newly established SA and WA algorithms include three parameters each. In contrast to DH and CH, SA and WA simulate reservoir release as a function of relative water storage in the reservoir and do not distinguish between irrigation and non-irrigation reservoirs, thus neglecting downstream water demand. The algorithms were calibrated against outflow, storage, storage anomaly, and estimated storage. For 35 out of the 100 reservoirs
with available observed inflow data, both observed and simulated inflows were used. Our analysis leads to the following conclusions:

- Whenever observations suitable for calibration are available, the parameters of the reservoir operation algorithm should be adjusted as we found the default parameter values  of the DH algorithm, particularly the irrigation reservoir parameter, are seldom the optimal parameter sets.
- Considering water demand in the modeling of irrigation and water supply reservoirs does not necessarily improve reservoir simulation compared to not considering water demand (modeling them the same as other reservoirs), potentially due to high uncertainty in demand estimation.



- Reservoir capacity, as accessed through GRanD datasets, may significantly differ from observed maximum daily storage values, which can lead to inaccurate reservoir simulations even with calibration. In such cases, comparing storage anomalies for model assessment is preferred, although this approach loses unique information about actual storage, which is typically the only available absolute surface water body storage in large-scale hydrological models.

- Calibration against outflow leads to relevant improvement only in simulated outflow and does not noticeably affect simulated storage, which remains very poor. However, using observed storage-related variables for calibration results in a clear improvement in storage simulation and a limited improvement in outflow simulation.

- To improve the poor performance of reservoir operation algorithms in storage simulation, they need to be calibrated against observed storage-related variables. However, even after calibration, their performance regarding storage during the validation phase remains worse than the model's performance regarding outflow.

- Among the three calibrated reservoir operation algorithms, the two newly introduced algorithms, WA and SA, performed better in storage simulation than CH. In general, SA performed slightly better than WA when using simulated inflow and calibration against storage anomalies.

- As currently available time series of absolute reservoir storage derived from remote sensing-based water storage anomalies often exhibit strong biases, and calibration against estimated storage did not outperform calibration against storage anomalies, we recommend implementing the SA reservoir operation algorithm in large-scale hydrological models. This approach can improve the quality of reservoir storage modeling after calibration against globally available, remote-sensing-based monthly time series of reservoir water storage anomalies.

- We found that using observed inflow instead of simulated inflow considerably improves the performance of the reservoir operation algorithms in terms of outflow simulation, but it does not have much impact on their performance in storage simulation.

- For most reservoirs, none of the three relatively simple reservoir operation algorithms can accurately represent the dynamics of both reservoir outflow and storage, even after calibration against observations of outflow or storage-related variables and even with observed inflow used in the simulation. The complexity of human decision-making cannot be captured by algorithms that rely solely on globally available information, even if their parameters are adjusted through calibration.

To improve large-scale hydrological modeling, we suggest leveraging recent and upcoming spaceborne information on reservoir water storage dynamics by implementing the SA reservoir operation algorithm, which enables calibration against storage anomalies. These algorithms are more suitable for large-scale applications than algorithms such as those of Chen et al. (2022) and Turner et al. (2021) that require daily inflow, storage, and outflow data — data that are rarely available outside the US. For reservoirs without data, the parameters can be regionalized from similar reservoirs that have data, or the current algorithm (DH) can be retained. Nevertheless, there is always potential to improve reservoir operation algorithms for better modeling of storage dynamics and outflow, and hybrid machine learning approaches, e.g. combining knowledge-based equations with deep learning, should be investigated for simulating reservoir dynamics. Improving the accuracy of inflow simulations and validating reservoir-related characteristics are considered more important than solely improving the algorithm itself. Finally, it should be noted that we applied a grid search method for calibration; we suggest using more advanced optimization methods to evaluate the impact of calibration approaches on the performance of reservoir operation algorithms..



*Code availability.* The WaterGAP 2.2e code is accessible through Müller Schmied et al. (2023b) and is licensed under the GNU Lesser General Public License version 3.

*Data availability.* All storage and outflow data obtained from different algorithms and calibration variants, as well as the calibrated parameters, are available in the supplement as Excel files. The reservoir characteristics are provided in Table S1. The observed data are available through Steyaert et al. (2022).

*Supplement.* The supplement related to this article is available online at [URL].

*Author contributions.* SMHM and PD designed the study. SMHM performed the modeling and wrote the first draft of the manuscript. PD contributed to the result analysis and editing of the paper. Both SMHM and PD were primarily responsible for writing the paper.

*Competing interests.* The authors declare that they have no conflict of interest.

*Acknowledgements.* This study was supported by funding from the German Research Foundation for the research unit "Understanding the global freshwater system by combining geodetic and remote sensing information with modeling using a calibration/data assimilation approach (GlobalCDA)". We acknowledge ChatGPT's assistance with editing certain sentences, while the authors have reviewed and refined the content and assume full responsibility for the publication.

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
