# Peer review of "The value of observed reservoir storage anomalies for improving the simulation of reservoir dynamics in large-scale hydrological models"

_Hydrology and Earth System Sciences, 2024_

## Author Response (AR1)

Dear Editor and Reviewers,

We sincerely thank the reviewers and the editor for their valuable comments and constructive suggestions, which have significantly contributed to enhancing the quality of our manuscript. We deeply appreciate the time and effort you have invested in reviewing our work. Below, we provide detailed, point-by-point responses in blue to each of the editor's and reviewers' comments. Changes made to the revised manuscript are clearly indicated in **bold**.

Thank you once again for your insightful feedback and guidance throughout this process.

Best regards,

The Authors
* * *
**Editor:** Dear Authors, thank you for your responses to the comments raised by two reviewers. Both reviewers consider the work presented as publishable, but have raised several comments, which you have now addressed through your responses. Following the recommendation of the reviewers, I would propose to now include the suggestions you have made in a revision of the manuscript. I will then seek the opinion of the reviewers to ensure the concerns raised have been adequately addressed. Please pay specific attention at the quality and clarity of presentation, as this was clearly raised as a concern.

_**Response:**_ Thank you for your thoughtful feedback and for considering our manuscript for further evaluation. We greatly appreciate the reviewers' constructive comments and are grateful for the opportunity to address their concerns. In accordance with your recommendation, we have carefully incorporated the suggested revisions into the manuscript. Particular attention has been given to improving the quality and clarity of the presentation, as this was emphasized as a key concern.
* * *
**Response to Anonymous Referee #1**

RC#1: This paper investigates whether storage anomalies, derived from remotely sensed data, can be used as data to enable the calibration of reservoir modules within large-scale hydrological models. This would enable improving the representation of reservoir within hydrological models for places where storage time series are not readily available (note this is never said explicitly in abstract or intro).

It uses 4 simple release rules (3 of which can be calibrated) from well-known global hydrological model WaterGAP, and 100 reservoirs from a comprehensive database of reservoirs from the United States, and for which observed storage is also available. It compares storage anomalies as the basis for calibration, with three other candidate time series: observed inflows, observed outflows, and estimated storage.

The idea is worthwhile, and the paper meets its main objective (TLDR: yes one can use storage anomalies). The beginning of Sections 4 and 5 suggest instead the focus was really on comparing reservoir release rules… but this in itself is a weaker contribution because there are many other release rules out there, why should we focus on these? To better reach the hydrological community beyond WaterGAP users, it would be best to instead show how the setup (different rules, observed vs. simulated inflows) shows that storage anomalies are a good choice of data, and the ability to get these for most reservoirs worldwide means rules presented in the paper can indeed be calibrated, and are of value (or in other words, the rules are basic and simple to calibrate, the existence of storage anomalies to carry out that calibration gives them value vs. the rest of the literature that they would not have otherwise). This will warrant rewriting quite a few bits.

_**Response:**_ We are pleased that you find the general idea of the paper worthwhile. We appreciate your feedback and have revised the text accordingly to align with your suggestions. Our primary focus is now on utilizing storage anomalies to fine-tune the reservoir algorithms. In the revised version, we have clarified why our focus is on using storage anomalies to calibrate reservoir modules and why we compared the selected reservoir algorithms. Notably, prior to calibration, we did not anticipate that the performance of the algorithms would not differ much in most cases. This highlights a common challenge for large-scale modelers: determining which algorithm is easiest to implement

while still achieving performance comparable to more complex approaches. Although calibration is required for the SA and WA algorithms, they are easier to implement than the calibrated H06 and even outperform it in general. For this reason, we believe that comparing simple release rules should be a substantial part of the manuscript. The study's aims have been reformulated as follows in the final paragraph of the introduction:

**"The main objective of this study is to investigate how monthly time series of observed reservoir-related data can improve the simulation of reservoir outflow and storage in continental or global hydrological models. We focus on the suitability of observed storage anomaly for calibrating reservoir release algorithms, as these anomalies can be obtained globally through remote sensing-based observations. We compare their informational value to that of scarcer outflow and absolute storage observations, as well as the simulation results achieved with an uncalibrated reservoir algorithm. We utilized in-situ storage and outflow data from the ResOpsUS dataset for 100 reservoirs in the US to calibrate three reservoir operation algorithms. All algorithms were implemented in the global hydrological model WaterGAP 2.2e (Müller Schmied et al., 2024). The parameters of the algorithms were estimated using as alternative calibration targets, 1) storage anomaly, 2) estimated storage (calculated based on storage anomaly and GRanD reservoir capacity, detailed in section 2.3), 3) storage, and 4) reservoir outflow. Calibration involved optimizing parameters individually for each reservoir, algorithm and calibration target. To explore, in addition, the sensitivity of the model results to the quality of the inflow data, we calibrated the algorithms for a subset of 35 reservoirs with available inflow measurements, using observed inflow instead of the inflow simulated by WaterGAP. Finally, for a subset of 21 reservoirs, we determined the effect of incorporating, in the case of irrigation and water supply reservoirs, the downstream water demand in the reservoir algorithms."**

In addition, we have completely reformulated the conclusions to focus on the applicability of (remotely sensed) reservoir storage anomalies in improving reservoir release algorithms.

**RC#1:** Key result: the key result really is whether storage anomalies fare well vs. the other data the reservoir model can be calibrated / validated against, under different conditions (e.g., observed or simulated inflows). However, this simple key statement is absent from the abstract, and this means the other statements on results seem disconnected from the purported aim.

***Response:*** We have revised the abstract accordingly and updated the manuscript text based on your previous comments. Below, please find the revised abstract:

**"Human-managed reservoirs alter water flows and storage, impacting the hydrological cycle. Modeling reservoir outflow and storage is challenging because it depends on human decisions, and there is limited access to data on reservoir inflows, outflows, storage, and operational rules. Consequently, large-scale hydrological models either exclude reservoir operations or use calibration-free algorithms for modeling reservoir dynamics. Nowadays, remotely-sensed information on reservoir storage anomalies is a potential resource for calibrating reservoir operation algorithms for a large number of globally distributed reservoirs. However, it is not yet clear what impact calibration against storage anomaly has on simulated reservoir outflow and absolute storage. In this study, we address this question using in-situ outflow and storage data from 100 reservoirs in the USA (ResOpsUS dataset) to calibrate three reservoir release algorithms, the well-established Hanasaki algorithm (CH) and two new storage-based algorithms, the Scaling algorithm (SA) and the Weighting algorithm (WA). These algorithms were implemented in the global hydrological model WaterGAP, with their parameters estimated individually for each reservoir and four alternative calibration targets: monthly time series of (1) storage anomaly, (2) estimated storage (calculated based on storage anomaly and GRanD reservoir capacity), (3) storage, and (4) outflow. The first two variables can be obtained from freely available global datasets, while the last two variables are not publicly available for most reservoirs worldwide. We found that calibration against outflow does not lead to skillful storage simulations in most reservoirs and improves the outflow simulations only slightly more than calibration against the three storage-related calibration targets. Compared to the results of the non-calibrated Hanasaki Algorithm (DH), calibration against both storage anomaly and estimated storage improved the storage simulation and slightly improved the outflow simulation. Calibration against storage anomaly resulted in 64 (39), 68 (45), and 66 (45) skillful storage simulations for CH, SA, and WA, respectively, during the calibration (validation) period, as compared to only 16 (15) for DH. Utilizing**

**estimated storage instead of storage anomaly does not provide added benefit, primarily due to inconsistencies in observed maximum water storage and storage capacity data from GRanD. Findings show that the default parameters of the Hanasaki algorithm rarely matched the calibrated parameters, highlighting the importance of calibration. Using observed instead of simulated inflow has a more significant effect on improving outflow simulation than calibration, whereas the opposite is true for storage simulation. Overall, the performance of the SA and WA algorithms is nearly equal, and both outperform the CH and DH algorithms. Moreover, incorporating downstream water demand into the reservoir algorithms does not necessarily improve modeling performance due to the high uncertainty in demand estimation. Therefore, to improve the modeling of reservoir storage and outflow in large-scale hydrological models, we recommend calibrating either the SA or the WA reservoir algorithm individually for each reservoir against remote sensing-based storage anomaly, unless in-situ storage data are available, and to improve reservoir inflow simulation.”**

**RC#1:** Description of methods. Several points there. First, there is no clear and concise explanation of how storage anomalies datasets are constructed. Similarly, it is never clear what the point of estimated storage is: it is constructed from monthly storage observations, is this something as readily available as storage anomalies? This should be added to 2.3 and 2.4, along with examples (maybe from the same 3 reservoirs from the results?) of how observed storage, storage anomalies and estimated storage compare for the U.S.

***Response:*** We had briefly mentioned the method for calculating storage anomalies in line 271 of the first version: *'For the storage anomaly, the mean storage during the calibration period is subtracted from the data.'* In the revised version, we have clarified this as follows:

**“Storage anomaly time series for each reservoir is calculated by subtracting the mean storage during the calibration period from the in-situ storage data for each reservoir.”**

In lines 274–284 of the first version, we had provided an explanation for the estimated storage calculations. Storage anomalies do not account for the bias term, which can lead to a calibrated absolute storage time series that deviates significantly from the actual absolute water storage. We evaluate the use of estimated storage instead of storage anomaly because it facilitates calibration against (an estimated values of) absolute storage rather than storage anomaly. Estimated storage can be calculated using either storage anomaly (from remotely sensed or in-situ data) or absolute storage, along with the water storage capacity of the reservoir. Since storage anomalies are available globally through remote sensing data, and the GRanD dataset includes capacity information for reservoirs, estimated storage can be easily calculated for each reservoir. To improve clarity, we have revised the text as follows [Lines 273-298]:

**“Using in-situ storage data, we derived two additional storage-related variables: the time series of storage anomaly and estimated storage. These variables can also be estimated using remote sensing data. Storage anomaly time series for each reservoir is calculated by subtracting the mean storage during the calibration period from the in-situ storage data for each reservoir. However, the storage anomaly lacks information about the bias term and calibrating against it can result in a simulated storage time series that significantly deviates from the observed water storage. Having actual absolute storage is advantageous, as reservoirs are the only surface water bodies for which we can model absolute storage within the WaterGAP. To provide an alternative, we calculated the “estimated storage time series”; this term refers to storage values that are not observed directly but are estimated using storage anomaly and the reservoir capacity C. First, we determined the storage changes time series by subtracting the initial month's storage anomaly value from the monthly storage anomaly values. Assuming the reservoir reaches maximum capacity at least once between 1980 and 2009, we calculated the maximum monthly storage change, termed Difmax. We then subtracted Difmax from the GRanD reservoir storage capacity to estimate the initial water storage for the first month. The estimated storage time series is then obtained by adding the storage changes to this estimated initial water storage. Since the data are monthly, and daily maximum storage is generally higher, we applied a 1.2 scaling factor to Difmax. This adjustment means that Difmax used in our calculations is 20% higher than the initially calculated value. This 20% increase is derived from the mean difference between the maximum daily storage and the monthly storage observed in 100 studied reservoirs (see Table S1). The calculation of estimated storage can be performed using either absolute storage or storage anomaly, as the time series of storage changes would remain the same in both cases. An example using GRanD ID 597 (Glen Canyon Dam, Lake Powell) clarifies the calculation of storage anomaly**

**and estimated storage. The mean observed storage value between 1980 and 2009 for Glen Canyon Dam is 22.45 km³. To obtain the storage anomaly time series for this reservoir, the value of 22.45 km³ is subtracted from all storage data for the reservoir over the entire period (1980–2019). For calculating estimated storage, the Difmax is 6.6 km³, which occurred in July 1983 (see Fig. S1). This is calculated as the storage anomaly value in July 1983 minus the initial storage anomaly value in January 1980. The initial storage is estimated as 25.1 km³ (the reservoir capacity reported by GRanD) minus 7.9 km³ (6.6 km³×1.2). This gives an initial storage value of approximately 17.2 km³. Storage changes are then added to the estimated initial storage to obtain the time series of estimated storage (Fig. S1c), e.g., the estimated storage for July 1983 is 23.8 km³, which is the sum of 17.2 km³ and 6.6 km³."**

**RC#1:** Second, the rationale for selecting 100 reservoirs is not super clear: why use geographical spacing on a 0.5 x 0.5-degree grid? That does not guarantee we have reservoirs that are not on upstream / downstream of one another.

***Response:*** The $0.5 \times 0.5$-degree grid is the resolution of the WaterGAP global hydrology model. We use this model because simulated streamflow data are needed, as observed inflow data are not available everywhere and in practice the reservoir operation algorithm should use simulated inflow for operation. Several filters were applied, resulting in the selection of only 100 reservoirs out of the 679 reservoirs listed in the ResOpsUS dataset, as outlined in Section "2.3 Data." The filters included:

1. A minimum of five years of storage and outflow data.
2. Storage capacity above 0.5 km³.
3. Ensuring only one reservoir per $0.5 \times 0.5$-degree grid (WaterGAP simulates all reservoirs within a single $0.5 \times 0.5$-degree grid as a single object).

Out of the 100 reservoirs, 51 are the most upstream reservoirs (see Table S1), meaning there are no global reservoirs upstream of them within the WaterGAP model. However, inflow into all reservoirs, regardless of their position, is based on the DH algorithm, where simulated inflow data by WaterGAP is first saved and then read from the saved files for all calibration experiments (as mentioned in lines 235–236 of the first version). This approach ensures that all calibrated algorithms use the same inflow data, preventing the operations of upstream reservoirs from affecting downstream reservoirs. We have clarified this further as follows [Lines 311-316]:

**"For comparison purposes, in all calibration experiments based on WaterGAP inflow, the inflow into reservoirs simulated by the DH algorithm was used to ensure that the same inflow data were applied across all algorithms. To achieve this, WaterGAP was first run with the DH algorithm to save the reservoir inflow data. These inflow data were then read from the saved files and used as the inflow source to model each reservoir independently. As a result, inflow into all reservoirs, regardless of their position, was based on the DH algorithm when applying the CH, SA, and WA algorithms, meaning that the operations of upstream reservoirs did not affect downstream reservoirs."**

**RC#1:** Third, the explanation for the release rule would warrant separate paragraphs / sub-sections for each. Things are abstract and quite difficult to follow as such. SA and WA should reference the original paper(s) that introduced them. To clarify, do these rules use demand estimates to adjust releases the way H06 does? I would also urge authors to better explain the practical difference between the rules, e.g., with diagrams showing release as a function of storage.

***Response:*** We have revised the text and added subsections for each reservoir algorithm. This paper introduces two new algorithms, SA and WA, for the first time; therefore, no prior references are available. These algorithms estimate releases without requiring water demand data. We have clarified this in the revised version as follows: [Lines 240-243]:

**"It should be noted that the new algorithms do not distinguish between irrigation and non-irrigation reservoirs; therefore, no water use data is required for their application, making their implementation easier than the H06 algorithm. This is because the estimation of downstream water demand at a large scale is generally very uncertain, and reservoirs are usually designed for multiple purposes."**

Additionally, We have included a diagram (Figure 1 in the revised version) to illustrate the release estimation process for each algorithm

**RC#1:** Four, a little bit more on calibration would be great. In practice, do you simulate N parameter sets and select the one with highest KGE? If so, how many parameter sets do you try? If not, what do you do?

*Response:* A more detailed explanation of the calibration process has been provided in section 2.4, "Model Variants and Calibration Approach." We considered N parameter sets for each algorithm, though the number of parameters varies across algorithms. For irrigation reservoirs, 5,800 parameter sets were used for the Hanasaki algorithm, while 8,000 parameter sets were applied for the SA and WA algorithms. Table S2 presents the number of parameter sets under different conditions for each algorithm, along with the range of each parameter. The following has be added to section 2.4 [Lines 300-306]:

**"The three reservoir operation algorithms were implemented in WaterGAP. For each algorithm, the algorithm-specific parameters ($a_1$, $a_2$, and $a_3$ for the CH, $p_1$, $p_2$, and $p_3$ for the SA and $q_1$, $q_2$, and $q_3$ for the WA) were estimated by optimizing the Kling–Gupta Efficiency (KGE) (Kling et al., 2012), including the trend term (see Eq. 10). This optimization was performed through a single-objective calibration against the monthly time series of four variables: outflow, storage, storage anomaly, and estimated storage (see Section 2.3). The parameters of each algorithm were calibrated using a grid search approach. Reservoir outflow and storage time series were simulated for all parameter sets listed in Table S2, and the parameter set corresponding to the highest KGE was selected."**

**RC#1:** Results. Several points to consider. SA vs. WA, the evidence for SA being better doesn't seem very robust, as (unless I have missed it) there's little evidence that the small nRMSE difference in favor of SA is statistically significant. I would instead, present SA and WA as equivalent throughout the paper (starting with the abstract).

*Response:* We agree with you that there is no definitive superiority of SA over WA. Our aim was to provide the reader with a clear choice, so we initially decided to conclude that, based on nRMSE, SA can be considered better than WA. However, based on your feedback, we have revised the text to indicate that SA and WA are equivalent.

**RC#1:** Figures 3 and 4: whilst it is great to see examples of reservoirs, the size of the figures and the choice of color / line style make this very difficult to read and understand. Please (1) use full page width (and even better, landscape page orientation) for each panel, and (2) adjust line width and line style, rather than using color codes that are not inclusive.

*Response:* Thank you for your feedback. In the revised version, we have plotted storage and outflow in separate figures, as suggested by the second reviewer, and adjusted the line width for the observations and the DH algorithm, improving clarity.

**Response to Anonymous Referee #2**

RC#2: The paper's goal is to improve reservoir representation in global hydrological models, specifically the WaterGAP model. It presents a comparison of different reservoir operation algorithms calibrated with different variants. One reservoir algorithm (DH, CH) uses the "Hanasaki" approach, and the other two use a new approach (SA, WA). Calibration is done using four different objective functions with storage anomaly from remote sensing data. I find the paper interesting. It does not show big improvements using the different reservoir algorithms and/or different objective functions, but it is solid work testing the hypothesis with different objective functions. I appreciate the summary of "findings" in the conclusion part.

The title "How can observational data be used to improve the modeling of human-managed reservoirs in large-scale hydrological models?" is a bit long and not to the point. First, I would remove the human-manage part. Maybe something like: Using observed data and reservoir operation algorithm to improve the representation of reservoirs in large-scale hydrological models.

*Response:* We appreciate your interest in our work. As the focus of the paper has shifted to the use of storage anomaly data for calibrating the reservoir algorithm, we have updated the title in the revised version to: " **The value of observed reservoir storage anomalies for improving the simulation of reservoir dynamics in large-scale hydrological models**".

RC#2: 2.3 Data. The data for storage, outflow, estimated outflow from ResOpsUS and GRanD is well described. But the description of the storage anomaly data from EO are lacking some information e.g some description of the SWOT mission, where the data is exactly from, and what are the postprocessing steps from SWOT to storage anomalies?

*Response:* We did not use data from the SWOT mission for calibration. The storage anomaly data were calculated based on observed in-situ storage by subtracting the mean over the calibration period. However, for discussion purposes, we briefly evaluated storage data from two remotely sensed sources: the Global Reservoir Storage (GRS) dataset introduced by Li et al. (2023) and the GloLakes dataset presented by Hou et al. (2024). Incorporating SWOT data is suggested as a potential avenue for future work. To clarify, we have moved the calculation of storage anomaly and estimated storage to the 'Data' section from the 'Model variants and calibration approach' section as follows [Lines 273-298]:

**"Using in-situ storage data, we derived two additional storage-related variables: the time series of storage anomaly and estimated storage. These variables can also be estimated using remote sensing data. Storage anomaly time series for each reservoir is calculated by subtracting the mean storage during the calibration period from the in-situ storage data for each reservoir. However, the storage anomaly lacks information about the bias term and calibrating against it can result in a simulated storage time series that significantly deviates from the observed water storage. Having actual absolute storage is advantageous, as reservoirs are the only surface water bodies for which we can model absolute storage within the WaterGAP. To provide an alternative, we calculated the "estimated storage time series"; this term refers to storage values that are not observed directly but are estimated using storage anomaly and the reservoir capacity C. First, we determined the storage changes time series by subtracting the initial month's storage anomaly value from the monthly storage anomaly values. Assuming the reservoir reaches maximum capacity at least once between 1980 and 2009, we calculated the maximum monthly storage change, termed Difmax. We then subtracted Difmax from the GRanD reservoir storage capacity to estimate the initial water storage for the first month. The estimated storage time series is then obtained by adding the storage changes to this estimated initial water storage. Since the data are monthly, and daily maximum storage is generally higher, we applied a 1.2 scaling factor to Difmax. This adjustment means that Difmax used in our calculations is 20% higher than the initially calculated value. This 20% increase is derived from the mean difference between the maximum daily storage and the monthly storage observed in 100 studied reservoirs (see Table S1). The calculation of estimated storage can be performed using either absolute storage or storage anomaly, as the time series of storage changes would remain the same in both cases. An example using GRanD ID 597 (Glen Canyon Dam, Lake Powell) clarifies the calculation of storage anomaly and estimated storage. The mean observed storage value between 1980 and 2009 for Glen Canyon Dam is 22.45 km³. To obtain the storage anomaly time series for this reservoir, the value of 22.45 km³ is subtracted from all storage data for the reservoir over the entire period (1980–2019). For calculating estimated storage, the Difmax**

**is 6.6 km³, which occurred in July 1983 (see Fig. S1). This is calculated as the storage anomaly value in July 1983 minus the initial storage anomaly value in January 1980. The initial storage is estimated as 25.1 km³ (the reservoir capacity reported by GRanD) minus 7.9 km³ (6.6 km³×1.2). This gives an initial storage value of approximately 17.2 km³. Storage changes are then added to the estimated initial storage to obtain the time series of estimated storage (Fig. S1c), e.g., the estimated storage for July 1983 is 23.8 km³, which is the sum of 17.2 km³ and 6.6 km³."**

**RC#2:** Calibration: The paper discusses outflow, storage, storage anomaly, estm. Storage as öbservations" to calibrate for with a trend including KGE. But which parameters are calibrated in each algorithm? How many parameters are calibrated? Is the number the same for all algorithms?

*Response:* The number of parameters for the SA and WA algorithms, as well as for the irrigation reservoirs in the Handaski algorithm, is the same: three parameters per reservoir in each algorithm. For the SA (WA) algorithm, the parameters correspond to three storage levels: storage > 70% capacity, storage between 40–70% capacity, and storage < 40% capacity. These parameters are denoted as p1, p2, p3 (q1, q2, q3) respectively. In the case of the Handaski algorithm, the parameters a1, a2 (specifically for irrigation reservoirs), and a3 are calibrated. However, the Handaski algorithm uses one parameter less for non-irrigation reservoirs, resulting in only two parameters per reservoir. The list of parameters and their respective ranges is presented in Table S2. We have modified the first paragraph of Section 2.4, 'Model Variants and Calibration Approach,' to include the following [Lines 300-304]:

**"The three reservoir operation algorithms were implemented in WaterGAP. For each algorithm, the algorithm-specific parameters ($a_1$, $a_2$, and $a_3$ for the CH, $p_1$, $p_2$, and $p_3$ for the SA and $q_1$, $q_2$, and $q_3$ for the WA) were estimated by optimizing the Kling–Gupta Efficiency (KGE) (Kling et al., 2012), including the trend term (see Eq. 10). This optimization was performed through a single-objective calibration against the monthly time series of four variables: outflow, storage, storage anomaly, and estimated storage (see Section 2.3)."**

**RC#2:** Line 187. Eq 6. You mentioned C in the equation before, but please put in storage capacity again (like in line 151)

*Response:* Done.

**RC#2:** Line 332 Fig 1: The part between 0 and 1 (or -0.73 and 1) is the interesting part. Maybe you can skip the values <-0.73 or sum them up or display them differently.

*Response:* Thank you for your suggestion. We have adjusted all values below -0.73 to -0.73 and regenerated Fig 1 (now Fig. 2) to better focus on values above -0.73.

**RC#2:** Fig 3/4: the calibration against outflow (b,d,f,h) is hard to distinguish. Maybe skip that part and put it in the appendix or show the differences to Observed.

*Response:* We created one plot for the storage time series and another for the outflow. The outflow time series plots (panels b, d, f, h) were moved to the supplementary material and enlarged to make them easier to distinguish.

---

## Author Response (AR2)

**Dear Editor,**

We have revised the manuscript based on the insightful comments you provided. All recommendations have been addressed in the updated manuscript. We would like to thank you for your thorough consideration and feedback, which helped us enhance our manuscript. Below, we have included a point-by-point response (in blue) to your comments. The suggested grammatical modifications are not listed here.
* * *
**Editor's comments**

**Editor:** Dear Authors, thank you for the careful revision of the manuscript following the reviewers' comments. The reviewers have reconsidered the manuscript and found that the concerns raised have been adequately addressed. I would like to propose the manuscript for publication. However, before proceeding, after a careful read of this final manuscript I believe there are some minor revisions that are required. Mostly these refer to the clarity of writing as well as some editorial corrections. I am attaching an annotated version of the manuscript. In this final revision, I would also ask to carefully review the results and discussion sections. Overal the results as presented and discussed are interesting and relevant, as also noted by the reviewers, but the writing it extremely dense and in many places difficult to follow. I would like to ask the authors to carefully review these sections and if the logic of message they aim to convey emerges clearly from the narrative. In the annotated attachment some examples are provided (as in the discussion), but the "dense" writing style pervades these sections. In many cases simplifying sentences with fewer sub-clauses can help a great deal.

**Response:** Thank you for your thoughtful feedback and for considering our manuscript for publication. We truly appreciate your effort to thoroughly review the manuscript and offer detailed comments to enhance its clarity and readability. We have carefully reviewed the annotated version of the manuscript and made revisions to address the editorial corrections and suggestions for enhancing the clarity of the Results and Discussion sections. Specifically:

- *Clarity and Simplification*: We have revisited the Results and Discussion sections to address your concerns.. Sentences with multiple sub-clauses have been simplified, and we have worked to ensure the logic and flow of the narrative are clear and concise.
- *Annotated attachment*: The specific examples highlighted in the annotated attachment have been addressed, and similar revisions have been applied consistently throughout the manuscript.
- *Final Review*: In addition to responding to the provided examples, we thoroughly reviewed the entire manuscript to ensure that the central messages emerge clearly and that the sections are accessible to a broad audience.

We hope these revisions meet your expectations and contribute to a more readable and compelling manuscript. If you require any further changes or clarifications, we will be happy to address them promptly.

Thank you again for your valuable feedback and for guiding our manuscript toward publication.

**Editor:** Line 330: This is not so clear. Maybe write this out a bit more clearly.

**Response:** We have revised this part as follows:

"In the case of the SA and WA approaches for considering downstream demand, the process involves using the provisional release $R_d'$ instead of $\bar{I}$ in Eqs. 7 and 8. Therefore, similar to the DH algorithm, Eq. 4 was used with the

default value for the parameter $a_2$ to estimate $R'_d$. Please note that, since the WA and SA approaches work with $\overline{I}$ and not $\overline{I'}$, $\overline{I}$ was applied in Eq. 4 instead of $\overline{I'}$ in the SA and WA approaches."

**Editor:** Line 460: sharp decline of what?

**Response:** It referred to the simulated storage. In the revised version, we have rephrased this section and removed the statement.

**Editor:** Line 464: So does this mean that the storage reported by GRAND is incorrect? If this is the case it is obvious the performance would poor! Somewhat trivial. Should that not have been corrected?

**Response:** Unfortunately, discrepancies exist in several cases between the reservoir storage capacities reported by GRanD and the maximum observed values, as also noted by Steyaert et al. (2022). We acknowledge that such discrepancies should ideally be addressed at the local scale. However, the primary goal of this paper is to provide guidance for large-scale models and their calibration against remotely sensed storage anomalies in regions where in-situ data is unavailable. Given that observed storage data is largely unavailable globally, and large-scale models heavily depend on GRanD data, we have opted not to modify the GRanD data. Instead, we aim to highlight one potential cause for the poor performance of reservoir operation algorithms.

**Editor:** Figure 4: The figure is already very busy. However, it may be useful to also indicate the relative storage lines (e.g. Srel = 0.4, 0.7 and 1.0) as these are of significance in the algorithms

**Response:** Thank you for your suggestion. The revisions have been made.

**Editor:** Table 3: This table is very difficult to read and understand if the results are significant. This is in part due to the improvements only being identified for skillful simulations, which means the total number considered varies (I assume this is the sum of the numbers in an outside the parenthesis). Numbers are often small so could all just be chance. Should a significance test be done?

**Response:** Indeed, the evaluated numbers are the same and equal to 21. However, to determine how often one approach yields better results, we first filter the 21 KGE values of that approach using a threshold of -0.73. Then, we compare whether considering or not considering water demand leads to a better outcome. Situations where considering or not considering water demand results in non-skillful simulations are disregarded and compared against the alternative variant (not considering or considering water demand, respectively). We believe in selecting models and parameters based on overall performance instead of relying on significance tests. We choose the model that typically delivers the highest performance, even if the second-best model shows similar results according to a significance test. When two models demonstrate the same performance, we prefer the simpler approach. In our case, since factoring in downstream water demand does not produce better results than ignoring it, we opt for the simpler model—i.e., the algorithm that does not account for water demand. Furthermore, we believe that the small sample size hinders our ability to conduct a reliable significance test.

We have revised the table caption for clarity as follows:

"**Table 3.** Comparison of reservoir simulation performance using different algorithms, both with and without considering downstream water demand for 21 irrigation and supply reservoirs. Numbers outside parentheses indicate the number of reservoirs (out of 21) where performance improves when downstream demand is taken into account.

In contrast, values inside parentheses represent reservoirs where ignoring downstream demand leads to higher KGE values. Improvements are noted only for skillful simulations achieving a KGE value greater than -0.73. All algorithms are calibrated against outflow, storage, storage anomaly, and estimated storage using KGE as the objective function. The inflow data is sourced from the WaterGAP model."

**Editor:** Line 630: It is not so clear that this sentence provides a summary of this sub-section. Please check. It comes across as somewhat confusing

**Response:** We have revised the summary sentence for this subsection to enhance clarity and better align with the key findings:

"In summary, our results suggest that enhancing the quality of inflow data is more crucial than calibrating reservoir operation algorithms, particularly when the objective is to achieve accurate outflow simulation. Only calibrating against storage anomalies does not ensure better outflow predictions."

**Editor:** Line 682: This is because for DH some defaults are provided. Can this not be done for WA and SA also?

**Response:** Both the WA and SA algorithms are not calibration-free; their parameters can be regionalized or adapted to other reservoirs using specific techniques that are beyond the scope of the current study. However, the key point is that calibration-free algorithms are generally not ideal for reservoir operation because each reservoir behaves differently. Nowadays, with storage anomaly data for reservoirs accessible through remote sensing, it is reasonable to use this information to calibrate reservoir operation algorithms for each reservoir individually. However, if such data is not available, traditional calibration-free algorithms remain the best available option.

**Editor:** Some of the comments in this final paragraph are more befitting of the discussion. For example, the discussion does not raise the issue of the grid search method - so whether that would improve results or change the outcomes is somewhat speculative. I would suggest a final paragraph that is better aligned to the scope of the study as a sort of overall conclusion.

**Response:** We have revised the final paragraph of the conclusions as follows:

"Although the algorithms introduced in this study outperform the conventional DH algorithm, there remains scope for improvement. For example, integrating knowledge-based equations with deep learning in hybrid machine learning methods could be beneficial for simulating reservoir dynamics. However, improving the accuracy of inflow simulations and validating reservoir-related characteristics is very likely more important for achieving better reservoir outflow and storage simulations than refining the algorithm itself."

**Editor:** Line 718: Is this intended to be a URL external to the article. If not then the supplementary material will be published with the article. The URL will be provided. Otherwise please provide the URL. The preferred approach is to publish this with the article on the HESS repository and not externally.

**Response:** The supplementary material will be published alongside the article.

**References**

Steyaert, J. C., Condon, L. E., Turner, S. W. D., and Voisin, N.: ResOpsUS, a dataset of historical reservoir operations in the contiguous United States, Sci. Data, 9 (1), 34, doi: 10.1038/s41597-022-01134-7, 2022.